# A generalized stress correction scheme for the MEB rheology: impact on the fracture angles and deformations

Mathieu Plante[1] and L. Bruno Tremblay[1]

[1]Department of Atmospheric and Oceanic Sciences, McGill University, Montréal, Québec, Canada

**Correspondence:** Mathieu Plante (mathieu.plante@mail.mcgill.ca)

**Abstract.**

The Maxwell Elasto-Brittle (MEB) rheology uses a damage parameterization to represent the brittle fracture of sea ice without involving plastic laws to constrain the sea-ice deformations. The conventional MEB damage parameterization is based on a correction of super-critical stresses that binds the simulated stress to the yield criterion but leads to a growth of errors in the stress field. A generalized damage parameterization is developed to reduce this error growth and to investigate the influence of the super-critical stress correction scheme on the simulated sea-ice fractures, deformations and orientation of Linear Kinematic Features (LKFs). A decohesive stress tensor is used to correct the super-critical stresses towards different points on the yield curve. The sensitivity of the simulated sea-ice fractures and deformations to the decohesive stress tensor is investigated in uniaxial compression experiments. Results show that the decohesive stress tensor influences the growth of residual errors associated with the correction of super-critical stresses, the orientation of the lines of fracture and the short-term deformation associated with the damage, but does not influence the long-term post-fracture sea-ice deformations. We show that when ice fractures, divergence first occurs while the elastic response is dominant, and convergence develops post-fracture in the long term when the viscous response dominates – contrary to laboratory experiment of granular flow and satellite imagery in the Arctic. The post-fracture deformations are shown to be dissociated from the fracture process itself, an important difference with classical Viscous Plastic (VP) models in which large deformations are governed by associative plastic laws. Using the generalized damage parameterization together with a stress correction path normal to the yield curve reduces the growth of errors sufficiently for the production of longer-term simulations, with the added benefit of bringing the simulated LKF intersection half-angles closer to observations (from $40 - 50°$ to $35 - 45°$, compared to $15 - 25°$ in observations).

## 1   Introduction

Sea ice is a thin layer of solid material that insulates the polar oceans from the cold atmosphere. When sea ice fractures and a lead opens, large heat and moisture fluxes take place between the ocean and the atmosphere, significantly affecting the polar meteorology on short time-scales and the climate system on long time-scales (Maykut, 1982; Ledley, 1988; Lüpkes et al., 2008; Li et al., 2020). The refreezing of leads contributes to the sea ice mass balance (Wilchinsky et al., 2015; Itkin et al., 2018); the associated brine rejection drives the thermohaline ocean circulation in the Arctic and vertical eddies in the ocean

mixed layer (Kozo, 1983; Matsumura and Hasumi, 2008). As such, the production of accurate seasonal-to-decadal projections using coupled models requires an accurate representation of sea-ice deformations along Linear Kinematic Features (LKFs).

As sea-ice models are moving to higher spatial resolutions, they become increasingly capable of resolving LKFs (Hutter et al., 2018, 2021). The representation of smaller-scale fracture physics on the other hand yet remains a challenge, as most sea-ice models are based on a continuum approach and rely on parameterizations to relate sea-ice deformations to unresolved
fractures. To this day, this is most commonly done using plastic rheologies or modifications thereof (Hibler, 1979; Hunke and Dukowicz, 1997), which have benefited from improved numerical schemes and efficiency to solve the highly non-linear momentum equation (Lemieux et al., 2008, 2014; Kimmritz et al., 2016; Koldunov et al., 2019). These models use plastic flow rules to represent the rate-invariance of sea-ice deformations at large spatio-temporal scales, in which the sea-ice can be considered ductile, but neglect the influence of the smaller-scale physics associated with the brittle fractures. A number of
other rheologies have been developed over the years to relate the sea-ice deformations to the smaller-scale fracture physics (Tremblay and Mysak, 1997; Wilchinsky and Feltham, 2004; Schreyer et al., 2006; Sulsky and Peterson, 2011; Rampal et al., 2016; Dansereau et al., 2016; Damsgaard et al., 2018). This brings a diversity of sea-ice rheologies, with different physical and numerical frameworks influencing the representation of sea-ice deformations at different scales.

The Sea Ice Rheology Experiment (SIREx, Bouchat et al., 2021; Hutter et al., 2021), a coordinated effort between several
ice-ocean modeling groups, assessed the pan-Arctic sea-ice deformation statistics simulated by different sea-ice rheologies. SIREx included the classical viscous-plastic (Hibler, 1979) and elastic-viscous-plastic (Hunke and Dukowicz, 1997) sea ice rheologies as well as the elastic-anisotropic (Wilchinsky and Feltham, 2004) and Maxwell-Elasto-Brittle (MEB, Dansereau et al., 2016) rheologies that include parameterizations of unresolved small scale physics. All participating sea ice models produced sea-ice deformation characteristics that have previously been associated with brittle behaviour, such as the scaling
and spatio-temporal coupling of sea-ice deformations (Bouchat et al., 2021), when run at sufficiently high resolution. The extent at which the inclusion of smaller-scale fracture physics improves this brittle behaviour thus remains an open question. Additionally, all rheologies produce similar angles between conjugate pairs of LKFs, a measure usually intimately related to the fracture mechanics and shear strength of a material (Bardet, 1991; Wang, 2007), showing a peek probability at 90° while the observed angles are in the range of 30-50° (Hutter et al., 2021). This calls for the improvement of sea-ice rheological
models, such as modifications of the mechanical strength parameters and yield curve (Bouchat and Tremblay, 2017; Ringeisen et al., 2019; Dansereau et al., 2019), the use of non-associated flow rules (in the case of classical plastic models, Ringeisen et al., 2021), or modifications of fine-scale fracture parameters (in the case of the EAP and MEB rheologies).

In the Maxwell Elasto-Brittle (MEB) rheology (Dansereau et al., 2016), the smaller-scale fracture physics is represented by a damage parameterization that was derived for rock mechanics and seismic models (Amitrano et al., 1999; Amitrano and
Helmstetter, 2006) and adapted for the large scale modelling of sea ice (Girard et al., 2011; Bouillon and Rampal, 2015; Rampal et al., 2016). This parameterization aims at representing the brittle character of sea-ice by using a damage parameter to represent the changes in material properties associated with fractures. This differs from parameterizations used in viscous plastic models in that the large-scale sea-ice deformations are not governed by plastic or granular flow rules. Instead, the sea-ice deformations in the MEB model are preconditioned by the presence of damage and the development of LKFs is associated with

the far-field stress concentration response to local damage, leading to the propagation of the damage (i.e. fractures) in space (Dansereau et al., 2019). While still based on the continuum assumption, it allows for brittle fractures to influence the sea-ice dynamics over shorter time-scales. It is currently used in the large-scale sea-ice finite element model neXtSIM (Rampal et al., 2019) and a finite difference version was recently implemented in the McGill Sea Ice Model Version 5 (McGill SIM5) (Plante et al., 2020).

The MEB rheology being relatively new, the extent to which the sea-ice deformations are sensitive to the numerical and material strength parameters has not been thoroughly tested yet. Nonetheless, the orientation of the simulated faults in uniaxial compression experiments are known to be sensitive to the angle of internal friction and to the Poisson ratio (Dansereau et al., 2019). This sensitivity is attributed to the influence of these parameters on the far-field stress concentration response to local damage, which determines the direction of the damage propagation. This suggests that the simulated angle of fracture may be

sensitive to the exact choice of damage parameterization, but has not yet been tested. Additionally, while the neXtSIM model performed well compared to other SIREx models, its different numerics (e.g. Lagrangian scheme with a triangular adaptive mesh) could also be responsible for the different scaling and localisation statistics (Bouchat et al., 2021). The finite difference implementation of the MEB rheology in the McGill SIM5 model, on the other hand, shows fast growth of residual errors at the grid scale – in ideal experiments – that significantly affect the post-fracture sea-ice deformations (Plante et al., 2020). These

errors result from the stress correction scheme used in the MEB rheology to define the growth of damage and to bring super-critical stresses back to the yield curve. To our knowledge, defining the damage in terms of the super-critical stress correction is new and unique to the EB and MEB sea-ice rheologies. For instance, many progressive damage models instead represent the damage parameter as a discrete function of the number of failure cycles (Amitrano and Helmstetter, 2006; Carrier et al., 2015). In continuum damage mechanics, the damage parameter is derived instead from thermodynamic laws (Murakami, 2012)

to simulate material fatigue. In the Elastic-Decohesive (ED) rheology, material damage is not parameterized but a decohesive strain rate explicitly represents the material discontinuity associated with the ice fracture and reduces the material strength of sea-ice, based on the orientation of the failure surface (Schreyer et al., 2006; Sulsky and Peterson, 2011).

In this paper, we present a generalization of the damage parameterization in which a decohesive stress tensor is introduced in the stress correction scheme such that the super-critical stresses can be brought back to the yield curve following different stress

correction paths in the stress invariant space. The generalization is used to reduce the growth of the residual errors associated with the stress correction and tested in uniaxial loading experiments to examine the influence of the stress correction on the simulated sea-ice fracture and deformations. The sensitivity of the simulated fracture angles to the decohesive stress tensor is also investigated to find the stress correction paths that present the added benefit of bringing the simulated fracture angles closer to observations.

This manuscript is organised as follows. In section 2, we present the MEB rheology and governing equations. The generalized stress correction scheme is described in section 3. The uniaxial loading experiment set-up is presented in section 4 along with the definition of diagnostics used to quantify the growth of damage and of residual errors. Results are presented in section 5, with a focus on the material behaviour in uniaxial compression experiments and its response to the changes in the damage

parameterization. In section 6, we provide a discussion on the generalized damage parameterization performance and other
model sensitivities. Conclusions are summarized in section 7.

## 2 Model

### 2.1 Momentum and continuity equations

The simulations are run using the MEB model implemented on an Eulerian, finite difference Arakawa C-grid in the McGill
SIM5 (Tremblay and Mysak, 1997; Lemieux et al., 2008; Plante et al., 2020). The vertically integrated 2D momentum equation
for sea ice can be written as (ignoring the sea surface tilt, the Coriolis and the ice grounding terms):

$$\rho_i h \frac{\partial \boldsymbol{u}}{\partial t} = \nabla \cdot \boldsymbol{\sigma} + \boldsymbol{\tau}, \tag{1}$$

where $\rho_i$ is the ice density, $h$ is the mean ice thickness, $\boldsymbol{u}$ $(= u\hat{\boldsymbol{i}} + v\hat{\boldsymbol{j}})$ is the ice velocity vector, $\boldsymbol{\sigma}$ is the vertically integrated
internal stress tensor and $\boldsymbol{\tau}$ is the net external surface stress from winds and ocean currents. This simplified formulation is
appropriate for short term uniaxial loading experiments but can result in small errors in ice velocity when using a realistic
model domain and forcing (Turnbull et al., 2017). Following Plante et al. (2020), we define the uniaxial loading by a surface
wind stress $\boldsymbol{\tau}_a$ and prescribe an ocean at rest below the ice:

$$\boldsymbol{\tau} \approx \boldsymbol{\tau}_a - \rho_w C_{dw} |\boldsymbol{u}| \boldsymbol{u}, \tag{2}$$

where $\rho_w$ is the water density, $C_{dw}$ is the water drag coefficient and $\boldsymbol{u}$ is the sea-ice velocity (see values in Table 1).

The prognostic equations for the mean ice thickness $h$ (volume per grid cell area) and concentration $A$ are written as:

$$\frac{\partial h}{\partial t} + \nabla \cdot (h\boldsymbol{u}) = 0, \tag{3}$$

$$\frac{\partial A}{\partial t} + \nabla \cdot (A\boldsymbol{u}) = 0, \tag{4}$$

where the thermodynamic source and sink terms are ignored.

### 2.2 Maxwell Elasto Brittle rheology

The MEB model differs from classical sea-ice models in that it represents the brittle character of sea ice using a damage
parameter to represent the effect of local fracture on the large-scale sea-ice material properties. The sea-ice deformations in the
MEB model thus occur post-fracture, rather than simultaneously as in most sea-ice model using granular or plastic flow laws,
and the formation of LKFs follows from the propagation of damage in space over short time-scales during the fracture process.

In the MEB rheology, the ice behaves as a visco-elastic material with a fast elastic response to forcing and a slower viscous
response that act over a longer-time scale. The governing equation for this visco-elastic material can be written as (Dansereau

et al., 2016, 2017; Plante et al., 2020):

$$\frac{\partial \boldsymbol{\sigma}}{\partial t} + \frac{1}{\lambda} \boldsymbol{\sigma} = E\mathbf{C} : \dot{\boldsymbol{\epsilon}}, \tag{5}$$

where $E$ is the elastic stiffness defined as the vertically integrated Young Modulus of sea ice, $\lambda$ is the viscous relaxation time-scale, $\mathbf{C}$ is the (fourth order) elastic tensor, ":" denotes the inner double tensor product and $\dot{\boldsymbol{\epsilon}}$ is the (second order) strain rate tensor. The tensors $\mathbf{C}$ and $\dot{\boldsymbol{\epsilon}}$ in the right hand side of Eq. 5 can be written in matrix form by representing the 3 independent components of the stress and strain tensors in a vector (see Rice, 2010), and the 9 independent components of the elastic modulus tensor in a 3x3 matrix, as:

$$\mathbf{C} = \frac{1}{1-\nu^2} \begin{pmatrix} 1 & \nu & 0 \\ \nu & 1 & 0 \\ 0 & 0 & 1-\nu \end{pmatrix}, \tag{6}$$

$$\begin{pmatrix} \dot{\epsilon}_{xx} \\ \dot{\epsilon}_{yy} \\ \dot{\epsilon}_{xy} \end{pmatrix} = \begin{pmatrix} \frac{\partial u}{\partial x} \\ \frac{\partial v}{\partial y} \\ \frac{1}{2}\left(\frac{\partial u}{\partial y} + \frac{\partial v}{\partial x}\right) \end{pmatrix} \tag{7}$$

where $\nu$ (= 0.33) is the Poisson ratio, which defines the relative amount of deformation on the plane parallel to the loading.

The relative importance of the elastic and viscous components (first and second terms on the left hand side in Eq. 5) are determined by the magnitude of the elastic modulus $E$ and viscous relaxation time-scale $\lambda$. $E$ and $\lambda$ are functions of the ice thickness, concentration and damage, such that the elastic term dominates when the ice is undamaged while the viscous term dominates when the ice is heavily fractured. The elastic modulus $E$ and viscous relaxation time-scale $\lambda$ are written as:

$$E = Yhe^{-a(1-A)}(1-d), \tag{8}$$

$$\lambda = \lambda_0(1-d)^{\alpha-1}, \tag{9}$$

where Y (= 1 GPa) is the Young Modulus of undeformed sea ice, $d$ is the damage parameter ($0 < d < 1$), $a$ (= 20) is the standard ice concentration parameter (Hibler, 1979; Rampal et al., 2016), $\lambda_0$ (= $10^5$s, $\approx$1 day) is the viscous relaxation time scale for undamaged sea ice and $\alpha$ is a parameter defining the post-fracture transition to the viscous regime. This damage-based transition to post-fracture viscosity represents a simplification of the observed plasticity (rate-independence) of sea-ice deformations (Coon et al., 1974; Tuhkuri and Lensu, 2002).

## 2.3 Yield criterion

Damage (or fracture) occurs when the internal stress state exceeds the Mohr-Coulomb failure criterion,

$$F(\sigma) = \sigma_{II} + \mu\sigma_I - c < 0, \tag{10}$$

where,

$$\sigma_I = \frac{\sigma_{xx} + \sigma_{yy}}{2}, \tag{11}$$

$$\sigma_{II} = \sqrt{\left(\frac{\sigma_{xx} - \sigma_{yy}}{2}\right)^2 + \sigma_{xy}^2}, \tag{12}$$

where $\sigma_I$ is the isotropic normal stress invariant (compression defined as negative), $\sigma_{II}$ is the maximum shear stress invariant, $(\sigma_{xx}, \sigma_{yy}, \sigma_{xy})$ are the components of the stress tensor, $\mu$ $(= \sin\phi)$ is the coefficient of internal friction of sea-ice, $\phi$ (= 45°) is the angle of internal friction, and $c$ is the vertically integrated cohesion, defined as,

$$c = c_0 h e^{-a(1-A)}, \tag{13}$$

where $c_0$ (= 10 kN m$^{-2}$) is the cohesion of sea ice derived from observations (Sodhi, 1997; Tremblay and Hakakian, 2006; Plante et al., 2020) or laboratory experiments (Timco and Weeks, 2010). No compressive or tensile strength cut-off are used in this analysis. The reader is referred to Table 1 for a list of default model parameters.

## 2.4 Damage parameterization

The prognostic equation for the damage parameter $d$ in the standard MEB rheology is parameterized using a relaxation term with time scale $T_d$ (= 1 s) as:

$$\frac{\partial d}{\partial t} = \frac{(1 - \Psi)(1 - d)}{T_d}, \tag{14}$$

where

$$\Psi = \frac{\boldsymbol{\sigma}_c}{\boldsymbol{\sigma}'} = \min\left(1, \frac{c}{\sigma'_{II} + \mu\sigma'_I}\right), \tag{15}$$

is a damage factor ($0 < \Psi < 1$), $\boldsymbol{\sigma}_c$ is the critical stress lying on the yield curve and $\boldsymbol{\sigma}'$ is the uncorrected stress state lying outside of the yield curve. Thermodynamic healing and the advection of damage are neglected as we are focusing on the ice fracture, which occurs at a timescale (seconds) much shorter than the healing and advection timescales (hours). Adding these terms does not change the results and conclusions presented in this paper but increases the localisation of the ice fractures with higher damage values that in turn increases ridging. These terms should be included in longer-term integration of the MEB model.

When the ice fractures, the damage factor $\Psi$ is used to scale the super-critical stresses back towards the yield curve. The prognostic equation for the temporal evolution of the super-critical stress tensor $\boldsymbol{\sigma}'$ is written as a relaxation equation of the same form as in Eq. 14:

$$\frac{\partial \boldsymbol{\sigma}'}{\partial t} = -\frac{(1 - \Psi)\boldsymbol{\sigma}'}{T_d}. \tag{16}$$

This stress correction scheme corresponds to scaling all the individual stress components by the factor $\Psi$, such that the stress state is corrected back onto the yield curve in the stress invariant space by following a line passing through the origin. This results in a dependency of the stress correction magnitude and of the damage on the super-critical stress state: i.e., the stress correction path becomes increasingly parallel to the yield curve for increasing compressive super-critical stresses, which also increases the numerical errors (Plante et al., 2020). We hereafter refer to this scheme as the "standard stress correction".

## 3 Generalized stress correction

We propose a generalized damage parameterization where the super-critical stresses are corrected back to the yield curve along a line oriented at any angle $\gamma$ from the y-axis in the stress invariant space (see Fig. 1). This generalization is developed with the goal of reducing the growth rate of the numerical errors in the MEB model by removing the dependency of the stress correction path on the super-critical stress state, while keeping the changes in the damage parameterization to a minimum so that it can be easily added to other MEB model implementations (and other damage-based models). In the MEB model, the exact path along which the super-critical stresses is returned to the yield curve is not known a priori, as the stress state never exceeds the yield criterion in reality. The proposed generalization allows to investigate the influence of the super-critical stress correction path angle on the simulated fractures and deformations. Other physically meaningful modifications of the stress correction that are based on thermodynamics principles are left for future work (see for instance Murakami, 2012) .

We define the damage factor in the generalized damage parameterization in terms of the shear stress invariant only, as:

$$\Psi = \frac{\sigma_{IIc}}{\sigma'_{II}}, \tag{17}$$

where $\sigma_{IIc}$ is the critical shear stress invariant. The equation defining the stress correction path with angle $\gamma$ (see Fig 1) can be written as:

$$\sigma_{II} = (1/\tan(\gamma))\sigma_I + B, \tag{18}$$

where $B \ (= \sigma'_{II} - 1/\tan(\gamma)\sigma'_I)$ is defined from the super-critical stress state ($\sigma'$). The critical shear stress invariant ($\sigma_{IIc}$) is then defined as the intersection point between the yield curve (Eq. 10) and the stress correction path (18),

$$\sigma_{IIc} = \frac{c + \mu\tan(\gamma)\sigma'_{II} - \mu\sigma'_I}{1 + \mu\tan(\gamma)}. \tag{19}$$

The damage factor can then be written in terms of the super-critical stress state invariants ($\sigma'_I, \sigma'_{II}$), the correction path angle $\gamma$ and the coefficient of internal friction $\mu$, as:

$$\Psi = \frac{c + \mu\tan(\gamma)\sigma'_{II} - \mu\sigma'_I}{(1 + \mu\tan(\gamma))\sigma'_{II}}. \tag{20}$$

In this manner, the correction of super-critical stresses can follow any path in the stress invariant space provided that the damage increases when ice fractures ($\Psi < 1$, or $\gamma < 90°$). This formulation can also be used with a yield curve with zero isotropic tensile strength (i.e. $c = 0$ kN m$^{-1}$), as opposed to the standard parameterization in which case any super-critical stress state is returned to the origin (see Eq. 15 when $c = 0$ N m$^{-1}$).

Note that using a stress correction path other than the standard path to the origin means that the corrected normal stress differs from the scaled super-critical stress $\Psi\sigma'_I$. We define this difference as the decohesive stress tensor (see Fig. 1), which is added to the damage parameterization to keep the corrected stress state on a given stress correction path. This effectively changes the stress correction while keeping the scalar definition of the damage parameter. The stress correction equation (Eq. 16) in the generalized damage parameterization then becomes,

$$\frac{\partial \boldsymbol{\sigma}'}{\partial t} = -\frac{(1 - \Psi)\boldsymbol{\sigma}' + \boldsymbol{\sigma}_D}{T_d}, \tag{21}$$

and the invariants of the decohesive stress tensor ($\sigma_{ID}, \sigma_{IID}$) are now defined as:

$$\sigma_{ID} = \sigma_{Ic} - \Psi\sigma'_I = \frac{c - \Psi(\sigma'_{II} - \mu\sigma'_I)}{\mu}, \tag{22}$$

$$\sigma_{IID} = 0, \text{ (by definition)}. \tag{23}$$

When $\tan\gamma = \sigma'_I/\sigma'_{II}$ and $\sigma_{ID} = \sigma_{IID} = 0$, we obtain the standard damage parameterization of Dansereau et al. (2016).

Note that the decohesive stress tensor used in this parameterization has a similar role as the decohesive strain rates used in the Elastic-Decohesive model (Schreyer et al., 2006). In Schreyer et al. (2006), the decohesive strain represents the discontinuity in sea-ice displacement associated with a fracture and relaxes the effective stress rates. It is derived from a decohesion function that depends on the mode of failure. Here, we do not define the strain discontinuity associated with the fractures, but use the decohesive stress tensor $\boldsymbol{\sigma}_D$ to prescribe the orientation at which the stress state is relaxed back onto the yield curve. This only indirectly influences the local strain rate via the constitutive equation.

## 3.1 Projected error

The error $\delta\Psi$ on the damage factor $\Psi(\sigma'_I, \sigma'_{II})$ can be written as (Plante et al., 2020):

$$\delta\Psi = \sqrt{\left(\frac{\partial\Psi}{\partial\sigma'_I}\right)^2 \delta\sigma'^2_I + \left(\frac{\partial\Psi}{\partial\sigma'_{II}}\right)^2 \delta\sigma'^2_{II}}, \tag{24}$$

where ($\delta\sigma'_I, \delta\sigma'_{II}$) are the errors of the calculated stress invariants. Using Eq. 21 and re-writing $\delta\sigma'_I$ and $\delta\sigma'_{II}$ in terms of the relative error $\epsilon$ (i.e., $\delta\sigma'_I = \epsilon\sigma'_I$, $\delta\sigma'_{II} = \epsilon\sigma'_{II}$), we obtain:

$$\delta\Psi = \sqrt{\frac{\mu^2}{(1+\mu\tan(\gamma))^2\sigma'^2_{II}}\epsilon^2\sigma'^2_I + \frac{(c-\mu\sigma'_I)^2}{(1+\mu\tan(\gamma))^2\sigma'^4_{II}}\epsilon^2\sigma'^2_{II}}, \tag{25}$$

$$= \Psi\epsilon\sqrt{\frac{\mu^2\sigma'^2_I + (c-\mu\sigma'_I)^2}{(c+\mu\tan(\gamma)\sigma'_{II} - \mu\sigma'_I)^2}}, \tag{26}$$

$$= \Psi\epsilon R \tag{27}$$

where R is the error amplification ratio.

Given that the uncorrected stress is close to the yield criterion (i.e. $\sigma'_{II} + \mu\sigma'_I - c \sim 0$), the error amplification ratio $R$ tends to infinity for,

$$\tan(\gamma) = -1/\mu, \tag{28}$$

which corresponds to a path that runs parallel to the yield curve. This result is consistent with the instabilities in the standard stress correction scheme during ridging reported in Plante et al. (2020), given that a line passing through the origin is nearly parallel to the Mohr-Coulomb yield curve for large compressive stresses. In contrast, the path that maximizes the denominator (smallest error growth) has $\gamma = 90°$. This path, however, corresponds to $\Psi = 1$ and does not create damage. The possible stress correction path angles $\gamma$ thus lie in the range $\arctan(-1/\mu) < \theta < 90°$.

Note that the error amplification ratio $R$ is small for $\sigma'_I < 0$, but becomes infinitely large at the yield curve tip when $\sigma'_{II}$ approaches 0 (see Eq. 25). This behaviour is opposite to that of the standard stress correction scheme, which has small $R$ values in tension and large values in compression (Plante et al., 2020). For this reason, we use both schemes (i.e. Eq. 20 in compression and Eq. 15 in tension, see Fig. 1b) and set the transition between the two schemes at the points where their paths are the same (i.e., at $\sigma'_I/\sigma'_{II} = \tan\gamma$, green line in Fig 1b). The damage factor is then defined as:

$$
\Psi = \begin{cases} \frac{c + \mu\gamma\sigma'_{II} - \mu\sigma'_I}{(1+\mu\gamma)\sigma'_{II}}, & \text{if } \sigma'_I < \sigma'_{II}\tan\gamma, \\ \frac{c}{\sigma'_{II} + \mu\sigma'_I}, & \text{otherwise.} \end{cases} \tag{29}
$$

## 4  Methods

### 4.1  Experiment setup

We test the numerical and material behaviour of the MEB model and the generalized damage parameterization in uniaxial compression experiments. Uniaxial experiments are designed to present conditions similar to those in laboratory experiments and have been used with MEB (Dansereau et al., 2016), VP (Ringeisen et al., 2019) and Discrete Element (Herman, 2016) models to assess ice fracture characteristics, LKF angles and intermittency. In this analysis, we use the experiment designed by Ringeisen et al. (2019) to test the sensitivity of the residual error growth, sea-ice deformation and LKF orientation on the correction path angle $\gamma$ in the generalized stress correction scheme. The model domain is 250 x 100 km with 1km spatial resolution. The initial conditions are 1m ice thickness and 100% concentration in the middle 60 km of the domain with two narrow bands of open water (20 km width) on each sides (Fig. 2). A solid-wall, Dirichlet boundary condition ($u = v = 0$) is used at the bottom, and open-water, Neumann boundary conditions ($\partial u/\partial n = 0$) are used on the top and sides. In all experiments, the forcing is specified by a downward surface stress $\tau_a$ (see Eq. 2) over the entire domain. This differs from Ringeisen et al. (2019) and Dansereau et al. (2016) where the upper boundary is represented by a moving wall acting as external forcing. The magnitude of $\tau_a$ is ramped up from 0 to 0.60 N/m$^2$ (corresponding to $\sim$20 m/s winds or $\sim$0.33 m/s surface currents) in a 2h period, and then remains constant.

Note that all simulations are performed without including heterogeneity in order to clearly identify the model performance (both numerics and physics), unless specified otherwise. This allows to quantify the growth of residual numerical errors in a problem with full symmetry and their impact on the simulated LKF orientation and post-fracture sea-ice deformations.

### 4.2  Numerical approaches

The MEB model is implemented in the McGill Sea Ice Model Version 5 (McGill SIM5) using an Eulerian, 2nd order finite difference numerical scheme (Tremblay and Mysak, 1997; Lemieux et al., 2014; Plante et al., 2020). The equations are discretized in space using an Arakawa C-grid and in time using a semi-implicit backward Euler scheme (Plante et al., 2020). A solution to the non-linear momentum and constitutive equations (Eqs. 1 and 5) is found using a Picard solver. The Picard solver uses an outer loop in which the equations are linearized and solved at each iteration using a preconditioned Flexible General

Minimum RESidual method (FGMRES, Lemieux et al., 2008). The non-linear terms are then updated and the linear problem solved again until the residual error $\epsilon_{res}$, defined as the L2-norm of the solution residual vector, is lower than $10^{-8}$ N/m$^2$ (Lemieux et al., 2014, for details). The prognostic equations for the tracers (Eq. 3, 4 and 14) are updated within the outer loop iteration using an IMplicit-EXplicit (IMEX) approach (Lemieux et al., 2014). The reader is referred to Plante et al. (2020) for more details.

### 4.3 Diagnostics

#### 4.3.1 Field asymmetry

We monitor the influence of the residual errors on the model solution in the simulations using a normalised domain-integrated asymmetry factor ($\epsilon_{asym}$) in the maximum shear stress invariant field ($\sigma_{II}$). This diagnostic measures the asymmetry in the model solution about the y-axis (the vertical center line) and represents a measure of the numerical accuracy given that the model equations, initial conditions and boundary conditions are all fully symmetric. The asymmetry factor is defined as:

$$\epsilon_{asym} = \frac{\sum_{i=a}^{b} \sum_{j=1}^{n_y} |(\sigma_{II})_{i,j} - (\sigma_{II})_{n_x - i, j}|}{\sum_{i=a}^{b} \sum_{j=1}^{n_y} |(\sigma_{II})_{i,j}|}, \tag{30}$$

where $(i,j)$ are the x-y grid indices respectively, $(n_x, n_y)$ are the number of grid cells in the x and y-directions and (a,b) are the indices of the first and last ice-covered grid cells on the x-axis.

Note that the field asymmetry measures the degradation of the originally fully symmetric problem as numerical errors are integrated, and includes the physical response to the integrated errors. This is in contrast with the residual error amplification ratio $R$, which is a measure of the local amplification of the residual error by the damage parameterization at a given time-step. The maximum $R$ values in the domain at each time-step ($R_{max}$) is also shown below to visualise the contribution of the damage parameterization to the growth of the residual errors.

#### 4.3.2 Damage activity

We quantify the development of fractures in the experiments using the damage activity $D$, defined as the total damage integrated over the original ice domain in a given time interval $\Delta$ (= 60 s):

$$D = \sum_{i=a}^{b} \sum_{j=1}^{n_y} \frac{d_{i,j}^{t+\Delta/2} - d_{i,j}^{t-\Delta/2}}{\Delta}. \tag{31}$$

This parameter is analogous to the damage rate in Dansereau et al. (2016, 2017) and is used to identify the time at which the ice fractures. Note that this definition of damage activity (or damage rate) emphasizes activity in undamaged ice (i.e. new fractures) and is not sensitive to activity in already heavily damaged ice.

#### 4.3.3 Fracture angle

The angles between conjugate LKFs in the Arctic are often discussed in relation with the orientation of the smaller-scale brittle fractures observed in laboratory under uniaxial compression loads (i.e., Marko and Thomson, 1977; Schulson, 2004). The

orientation of such compressive-shear fractures is often related to brittle fracture theories (e.g. to the development of wing cracks, Schulson, 2004; Wachter et al., 2009) and in terms of granular properties such as Coulombic friction or dilatancy (Erlingsson, 1988; Tremblay and Mysak, 1997; Overland et al., 1998).

Here, we define the fracture angle $\theta$ as the angle between the y-axis and the fracture lines (see Fig. 2), and compare the simulated fracture angles in our experiments to two theories that are often used to describe the orientation of fractures: the Mohr-Coulomb fracture theory and the Roscoe theory of dilatancy. Widely used in geoscience and engineering, the Mohr-Coulomb theory (Coulomb, 1773; Mohr, 1900) relates the orientation of fractures to the angle of internal friction, as:

$$\theta = \frac{\pi}{4} - \frac{\phi}{2}. \tag{32}$$

In the Roscoe theory (Roscoe, 1970), the fracture angle is defined instead in terms of the angle of dilatancy ($\delta$) of the granular material:

$$\theta = \frac{\pi}{4} - \frac{\delta}{2}. \tag{33}$$

If $\delta = \phi$, the two theories give the same fracture angle $\theta$. In general, the fracture angle in geomaterial and soils falls between values predicted by the Mohr-Coulomb and Roscoe theories with zero dilatancy ($\delta = 0$) (Arthur et al., 1977; Bardet, 1991).

In our experiment, the fracture angle is calculated graphically for each individual simulation. We define the uncertainty as $\pm \tan(W/L) \sim \pm 2°$, where $W$ is the fracture width (typically a few grid cells wide, or $\sim$ 2-5 km) and $L$ is the fracture length ($\sim$ 45 km). This error increases to $\pm 6°$ for the few cases where the fracture is not as localized.

## 5 Results

### 5.1 Control simulation: standard damage parameterization

In the control simulation, a pair of conjugate LKFs first appear when the surface forcing $\tau_a = 0.29$ N m$^{-2}$, along with secondary lines that are the results of interactions between the ice floe and the solid boundary that extends across the full width of the domain at the base (Fig. 3). All LKFs are oriented at 39° from the y-axis, smaller than reported by Dansereau et al. (2019) using a Finite Element implementation of the same model ($\theta = \sim 43°$) and higher than seen in observations ($\theta = \sim 15\text{-}25°$ Marko and Thomson, 1977; Hibler III and Schulson, 2000; Schulson, 2004; Hutter et al., 2021). This orientation also falls in between that predicted by the Mohr-Coulomb ($\theta = 22.5°$) and Roscoe theories ($\theta = 45°$ when $\delta = 0$), in accord with the common observation that both the angle of internal friction and the dilatancy ($\delta$) are important in defining the fault orientation (Arthur et al., 1977; Vardoulakis, 1980; Balendran and Nemat-Nasser, 1993).

The deformation along the fully developed LKFs in our experiment is mostly shear and convergence (i.e. ridging, Fig. 3c-d). This contrasts with the early stage of the LKF development during which the material response to the new damage is elastic and shows mostly divergent deformations (see the positive strain rates in Fig. 4b). This elastic response to damage influences the propagation of the fractures in space at short time-scales (seconds) governed by the elastic waves speed. The convergent deformations only develops over a longer time-scales as the sea-ice deformation continues post-fracture in the damaged ice and

the deformation transitions from the elastic- to the viscous-dominated regime. This transition is clearly seen in the development of a linear dependence between stress and strain-rate invariants (scaled by $(1-d)^3$), where the slope corresponds to the viscosity (see the transition from 4 b,d, to f). The simulation reaches steady state with deformations that are fully viscous and localized in the heaviest damage areas (Fig. 4e-f). This causes a predominance of shear and convergence deformation along the LKFs
throughout the simulation.

The asymmetries in the solution are very small at the beginning of the simulation (t $\leq$ 57 min), and do not grow until fractures occur (Fig. 5a-b). As the LKFs develop, small errors grow rapidly with $\epsilon_{asym}$ increasing in large steps crossing multiple orders of magnitude. Note that the model is always iterated to convergence with a strict residual error tolerance ($\epsilon_{res} = 10^{-8}$ N m$^{-2}$). The steep growth in $\epsilon_{asym}$ is associated with large ($> 1$) values of the error amplification ratio $R$ (see Eq. 27), which reach
$\sim$20 in the control simulation (Fig. 5b). Since $\epsilon_{asym}$ is a domain-integrated quantity, it increases in time following large local error growths $R$. This illustrates the long-range and long-term influence of residual errors, which act on the development of the future fractures. Note that $\epsilon_{asym}$ saturates when the $\sigma_{II}$ field is no longer symmetric, and becomes insensitive to additional error growth. We assess the precision of the solution using the maximum error amplification ratio $R_{max}$, which indicates the level of amplification of residual errors in the simulations, at times by more than one order of magnitude locally ($R_{max} > 10$).

**5.2 Generalized stress correction**

The generalized damage parameterization reduces the growth of residual errors, with decreasing asymmetry factor and maximum error amplification ratio $R_{max}$ for increasing path angle $\gamma$ (Fig. 6). In particular, using $\gamma > 0°$ stabilises the damage parameterization and eliminate the large spikes in $R_{max}$ seen in the control simulation or when using $\gamma < 0°$, where the amplification ratio $R$ increases by up to two orders of magnitude locally (Fig. 6b). The increased stability results in an overall
smaller and smoother growth of the asymmetry factor $\epsilon_{asym}$ (Fig. 6a), allowing for longer-term symmetrical simulations that include post-fracture deformations. Note that despite this improvement, the asymmetry factor $\epsilon_{asym}$ still grows over time as the simulations remain sensitive to the residual errors in heavily damaged ice, due to the non-linear relationship between the sea ice deformation and the damage. This effect is less important when using large correction path angles ($\gamma > 45°$) due to a slower LKF development, as discussed below.

Results show that the LKF orientation is sensitive to the decohesive stress tensor, with decreasing angle $\theta$ for increasing stress correction path angle $\gamma$ (Fig. 7). This finding is in line with results from Dansereau et al. (2019), where the orientation of faults was related to the far-field stress associated with the collective damage. In the MEB model, the far-field stresses directly depend on the corrected stress state, which includes $\boldsymbol{\sigma}_D$ in the generalized damage parameterization. Increasing the correction path angle $\gamma$ reduces the LKF angles, in better agreement to observations.

The correction path angle $\gamma$ influences the time-integration required to reach the same damage and deformation rates (Fig. 8) along the LKFs. This is due to the fact that increasing the angle $\gamma$ reduces the amount of damage for the same super-critical stress state because the stress correction path approaches the horizontal and $\Psi$ is closer to 1. The simulated ice deformations are otherwise mostly insensitive to the correction path angle; i.e. all simulations have divergence during the initial elastic response when the ice fractures followed by a transition to viscous deformations where shear and convergence deformations

are predominant (Fig. 8a). In contrast with plastic flow (Ringeisen et al., 2019, 2021) or typical granular material behaviour (Balendran and Nemat-Nasser, 1993; Tremblay and Mysak, 1997), divergent post-fracture deformation is only present when tensile stresses develop, e.g. at the intersection between conjugate LKFs. This behaviour stems from the use of post-fracture viscosity to represent the large-scale sea-ice deformations, and differs from classical VP model, which represent the observed plasticity of sea-ice deformations at the macro-scale (Coon et al., 1974; Tuhkuri and Lensu, 2002) but do not represent the

brittle component of the fractures nor discontinuities in material properties.

## 5.3   Sensitivity to $\phi$ and $\nu$

Repeating the experiment using different angles of internal friction ($\phi$) shows that the LKF orientations decrease with increasing $\phi$. The simulated angles $\theta$ fall within the envelope from the Mohr-Coulomb and Roscoe theories, except for small angles of internal friction ($\phi < 20°$), a value that is rarely observed for granular materials (Fig. 9). Note that the sensitivity of the

LKF orientation to the coefficient of internal friction also disappears for small angles of internal friction ($\phi < 20°$) when using a large correction path angle ($\gamma = 60°$ in Fig. 7). When both the stress correction path and the yield criterion approach the horizontal, fracture yields large stress corrections but small damage increases (i.e., $\Psi = 1$), such that the LKF orientation is mostly governed by the stress correction and weakly sensitive to other model parameters. Based on these results, we suggest the use of a correction path that is normal to the yield criterion ($\gamma = \arctan \mu$, see black points in Fig. 9).

Decreasing the angle of internal friction reduces the shear strength of sea ice for a given normal stress, such that the fracture develops earlier in the simulation (i.e. under smaller surface forcing, Fig. 10). It also reduces the divergence associated with the elastic response when ice fractures and increases the convergence in the post-fracture viscous regime. This result is typical for granular material, with smaller fault orientations (larger angles of internal friction) associated with larger angles of dilatancy Bolton (e.g. the sawtooth model, 1986)).

The orientation of LKFs is not sensitive to the Poisson ratio when the generalized stress correction scheme is used with a fixed stress correction path angle $\gamma$ (Fig 11). This is in contrast with simulations using the standard stress correction scheme, where the fracture angle decreases with increasing $\nu$ (see blue points in Fig. 11, and Dansereau et al., 2019). Note that the Poisson ratio also affects the amount of shear and normal stress concentration associated with a local discontinuity in material properties (Karimi and Barrat, 2018). The fact that the LKF orientation is not affected by the changes in Poisson ratio thus

indicates that the stress concentration and propagation of the fracture in space is mainly controlled by the stress correction rather than by the relaxation of material properties with damage. We speculate that the sensitivity of the LKF orientation to the Poisson ratio in the standard stress correction scheme stems from the dependency of the stress correction path angle to the super-critical stress state (i.e. $\gamma = tan^{-1}(\sigma'_I / \sigma'_{II})$).

## 6   Discussion

The results presented above show that the generalized stress correction scheme reduces the growth of the residual error associated with the damage parameterization. Despite the improvement, some asymmetries are still present in the simulations

($\epsilon_{asym} < 10^{-2}$). This is due to the memory in the damage parameter (i.e. an integrated quantity) where residual errors accumulate and influence the temporal evolution of the solution. In regions of heavily damaged ice, the integrated errors in the damage parameter result in large errors in the stress state due to the cubic dependence of the Maxwell viscosity $\eta$ on $d$ (Eq. 9). Future work includes replacing this formulation with a function that decreases the sensitivity of the Maxwell viscosity $\eta$ for small changes in $d$ around $d = 1$.

Overall, the use of a decohesive stress tensor yields smaller simulated LKF angles, without significantly impacting the material deformations. Using a large correction path angle $\gamma$ ($> 45°$), however, significantly slows the damage production and reduces the simulated sensitivity of the LKF orientation to the mechanical strength parameters. Based on these results, we suggest using a correction path that is normal to the yield criterion ($\gamma = \arctan \mu$). This value brings the simulated LKF angles closer to observations (see black points in Fig. 9) and reduces the amplification of residual errors, while correcting the super-critical stresses towards the closest point on the yield curve. Our implementation thus represents a generalization of the damage parameterization that can be easily implemented numerically and used to improve the performance of MEB models. Whether these improvements are also seen in the context of pan-Arctic simulations remains to be tested, and is the subject of future work.

The simulation results show that in the MEB model, the damage develops at short time scales during which the elastic component of the rheology is important, while most of the deformations occur post-fracture over a longer time scale in the heavily damaged ice. This is in contrast with plastic models, in which a flow rule simultaneously dictates both the LKF development and the relative amount of shear and normal deformations occurring along the LKFs. The decoupling between the development of damage and the post-fracture deformations in the MEB model explains that the type of deformations in the LKFs remains similar (uniaxial convergence, i.e. ridging, contrary to observation, Stern et al., 1995) despite the use of different stress correction path $\gamma$. This behaviour stems from the dominance of the viscous regime post-fracture: lead opening cannot occur when the stress state is compressive and remains limited to locations where tensile stresses are present, such as at the intersection of the LKFs. This is contrary to granular theories, in which the distribution of contact normals determines the amount of ridging or lead opening (i.e. dilatancy) that is occurring when forced in uniaxial compression (Balendran and Nemat-Nasser, 1993). This indicates that the decohesive stress tensor cannot be used to influence the deformations associated to the fracture of ice in the MEB rheology unless other parameterizations, such as including a decohesive strain tensor during the fractures (e.g., see Schreyer et al., 2006; Sulsky and Peterson, 2011), are added to the rheology.

The viscous dissipation timescale ($\lambda$) in our model is set based on observations ($\sim 10^5$, Tabata, 1955; Hata and Tremblay, 2015), and is one order of magnitude smaller than in other MEB implementations (Dansereau et al., 2016; Rampal et al., 2019). The results from the model are robust with respect to the exact value of $\lambda$ for a range $10^5 - 10^7$; the increased $\lambda$ being compensated by larger damage values along the LKFs. For even larger $\lambda$ values, divergent deformations persist longer in the simulation and the transition from elastic- to viscous-dominated regime occurs later in the simulation (see Fig. 12), decreasing the overall convergence along the LKFs. If the transition to the viscous regime is removed (e.g. by setting $\alpha = 1$), divergence dominates throughout the simulations and reaches large values as the leads open. The elastic waves, however, are no-longer dissipated in the LKFs, leading to large and noisy deformation fields (divergence/convergence). These findings call

for a different viscosity-dependence on damage leading to both dissipation of elastic waves and a more realistic post-fracture deformation field.

Note that the results presented above were presented using a single space and time resolution, ice sample aspect ratio and without using heterogeneity. While the exact localisation of the LKFs in the simulations is affected by these parameters, the overall physics and sensitivity to the damage parameterization are robust to these changes. For instance, repeating the experiment by doubling the space resolution or the width of the ice sample does not change the LKF position and orientation (not shown). On the other hand, adding heterogeneity changes the LKF development by forming irregular sliding planes instead of the linear diamond shapes (Fig. 13a), naturally creating contact points where ridging occurs with lead opening elsewhere along the LKFs. This effectively creates a form of dilatancy typical of granular materials (see alternating divergence and convergence in Fig. 13c) and leads to the formation of many secondary fractures, but the overall LKF orientations and their sensitivities otherwise remain the same as presented in this manuscript. Heterogeneity was also documented to be responsible for the localisation and intermittency of the sea-ice fractures, properties that are not investigated in our manuscript. These properties and their sensitivity to the decohesive stress tensor and other physical or numerical parameters requires more investigation and is the subject of future work.

## 7 Conclusion

We propose a generalized stress correction scheme for the damage parameterization to reduce the growth of residual errors in the MEB sea ice model documented in (Plante et al., 2020). To this end, we scale the damage factor $\Psi$ based on the super-critical maximum shear stress invariant ($\sigma'_{II}$) only, together with a decohesive stress tensor defining the path from the super-critical stress state to the yield curve. With this added flexibility to the choice of stress correction path, we determine the influence of the super-critical stress correction on the simulated sea-ice deformations and LKF orientation in the context of uniaxial compression experiments similar to those presented in Ringeisen et al. (2019). This knowledge will serve as a basis for the development of other components to the damage parameterization to improve the simulated sea-ice deformations.

Our results show that in the MEB rheology, most of the deformations occur post-fracture in heavily damaged ice, where the viscous term is dominant. This causes a predominance of convergence (ridging) in the LKFs, contrary to laboratory experiments of granular materials and satellite observations of sea ice. The use of a decohesive stress tensor influences the LKF orientation in the sea ice cover, but does not influence the type of deformation rates (convergence and shear), nor the simulated dilatancy. Future work will involve the modification of the non-linear relationship between the Maxwell viscosity and the damage. We also show that the sensitivity of the LKF orientation to the Poisson ratio, seen when using the standard damage parameterization, disappears when using the generalized stress correction scheme with a fixed stress correction path. This suggests that in the MEB model, the stress concentration and fracture propagation is governed by the stress correction rather than by the relaxation of the mechanical properties associated with the damage.

Based on our results, using the generalized damage parameterization with a stress correction path normal to the yield curve reduces the growth of residual errors and allows longer term simulations with post-fracture deformations. Using this stress

correction path also reduces the orientation of LKFs by $\sim 5°$, bringing them closer to observations. Despite these improvements, some error growth remains inherent to the formulation of the damage parameterization. Whether this might be improved by removing the dependency of the damage parameters on the damage factor (and on the super-critical stress state) will be explored in future work.

*Code availability.* Our sea-ice model code and outputs are available upon request.

*Author contributions.* M. Plante coded the model, ran all the simulations, analyzed results and led the writing of the manuscript. B. Tremblay participated in regular discussions during the course of the work and edited the manuscript.

*Competing interests.* The authors declare that they have no conflict of interest.

*Acknowledgements.* We are grateful to the Fonds de recherche du Québec – Nature et technologies (FRQNT) for financial support to Mathieu Plante during the course of this work as well as to the Natural Science and Engineering and Research Council (NSERC) Discovery Program

and the Environment and Climate Change Canada Grant & Contribution for grants awarded to Bruno Tremblay.

This work is a contribution to the research program of Québec-Océan and to the ArcTrain International Training Program. We thank the three anonymous reviewers for their useful comments and suggestions during the open discussion process. We also thank Amélie Bouchat, Damien Ringeisen, Martin Losch and Jean-François Lemieux for useful discussions during the implementation of the MEB model and the generalised stress correction.

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

**Table 1.** Default Model Parameters

| Parameter | Definition | Value |
| --- | --- | --- |
| $\Delta x$ | Spatial resolution | 1 km |
| $\Delta t$ | Time step | 0.2 s |
| $T_d$ | Damage time scale | 1 s |
| Y | Young Modulus | $10^9$ N m$^{-2}$ |
| $\nu$ | Poisson ratio | 0.33 |
| $\lambda_0$ | Viscous relaxation time | $10^5$ s |
| $\alpha$ | Viscous transition parameter | 3 |
| $\phi$ | Angle of internal friction | 45° |
| $c_0$ | Cohesion | 10 N m$^{-2}$ |
| $\rho_a$ | Air density | 1.3 kg m$^{-3}$ |
| $\rho_i$ | Sea ice density | $9.0 \times 10^2$ kg m$^{-3}$ |
| $\rho_w$ | Sea water density | $1.026 \times 10^3$ kg m$^{-3}$ |
| $C_{da}$ | Air drag coefficient | $1.2 \times 10^{-3}$ |
| $C_{dw}$ | Water drag coefficient | $5.5 \times 10^{-3}$ |

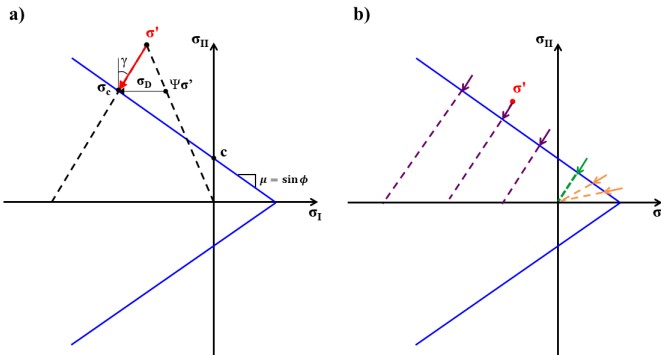

**Figure 1.** a) Mohr-Coulomb yield criterion ($\pm\sigma_{II} + \mu\sigma_I = c$, blue lines) in stress invariant space. $\boldsymbol{\sigma}'$ is the uncorrected super-critical stress state, $\boldsymbol{\sigma}_c$ the critical stress state for a given correction path angle $\gamma$ (red dashed line) and $c$ is the cohesion. The decohesive stress tensor $\boldsymbol{\sigma}_D$ is defined as the difference between $\boldsymbol{\sigma}_c$ and the scaled super-critical stress ($\Psi\boldsymbol{\sigma}'$). b) Proposed correction paths for various super-critical stresses $\boldsymbol{\sigma}'$ that minimizes the error amplification ratio (R), which consist of the standard parameterization for large tensile stresses (orange) and a correction path with $\gamma = 45°$ for small tensile and compressive stresses (purple). The green line indicates the transition between the two formulations.

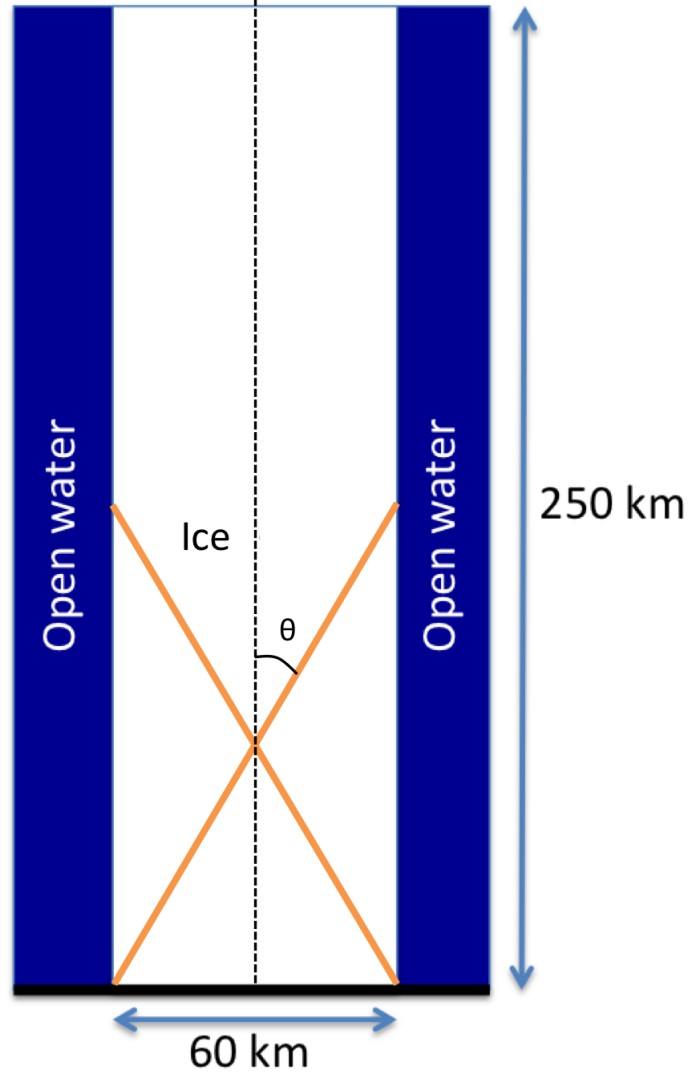

**Figure 2.** Idealized domain for uniaxial compression simulations, with a solid boundary (Dirichlet conditions, $u = v = 0$) at the bottom, and open boundaries (Neumann conditions, $\partial u / \partial n = 0$) on the sides and top. The initial conditions are h = 1m and A = 100% in a region of 250 x 60 km in the center of the domain (white), with two 20 km wide bands of open water on each side (blue). The orientation of the LKFs ($\theta$) is defined as half of the angle between conjugate pairs of fracture lines (orange lines).

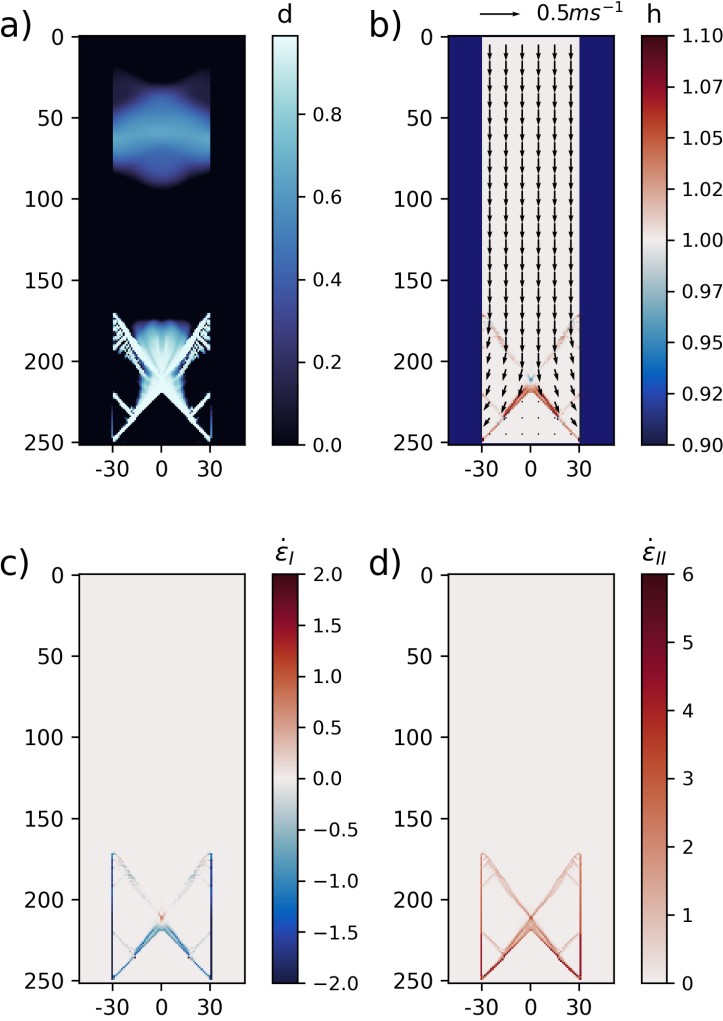

**Figure 3.** a) Damage (unitless), b) ice thickness (m, color) and velocity vectors (m s$^{-1}$), c) mean normal strain rate invariant ($\dot{\epsilon}_I$, day$^{-1}$) and d) maximum shear strain rate invariant ($\dot{\epsilon}_{II}$, days$^{-1}$), after two hours of integration in the control simulation using the standard stress correction scheme.

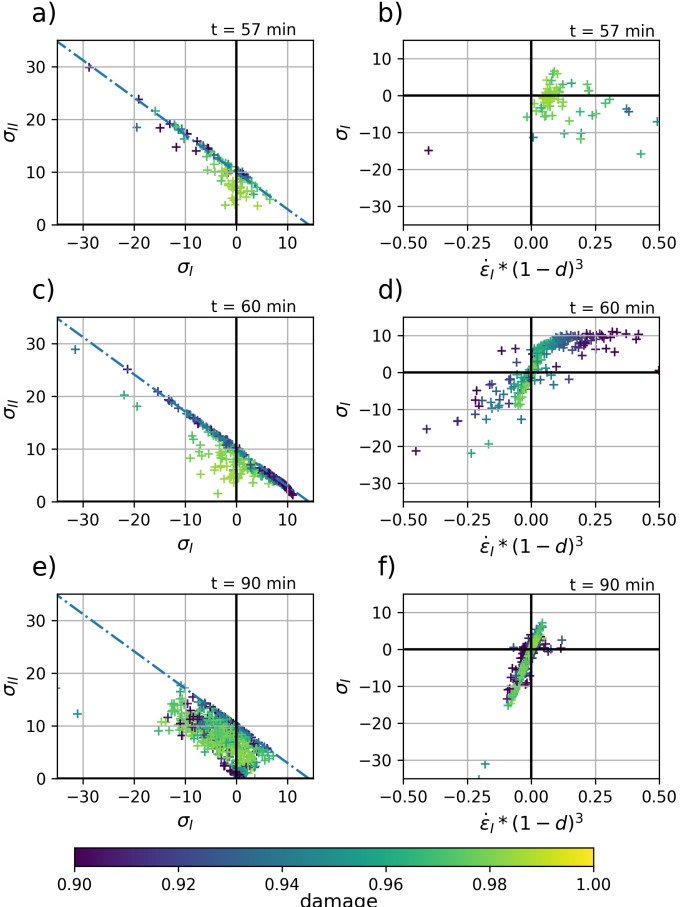

**Figure 4.** Scatter plots of local stress invariants ($\sigma_I$ vs. $\sigma_{II}$, in kN m$^{-1}$, left column), normal stresses and scaled strain rate invariants ($\sigma_I$ vs. $(1-d)^3\dot{\epsilon}_{II}$, right column) in heavily damaged (d > 0.9) grid cells, at t = 57 min (during the fracture development, top row), t = 60 min (a few minutes after the fracture, middle row), and t = 90 min ($\sim$ 30 min after the fracture, bottom row). Color indicates the local damage. The strain rates are normalised to account for the non-linear dependency of the viscosity $\eta$ on the damage parameter. The gradual alignment of the points in the $\sigma_I$ vs. $(1-d)^3\dot{\epsilon}_{II}$ diagram indicate the development of a linear-viscous stress-strain relationship over time.

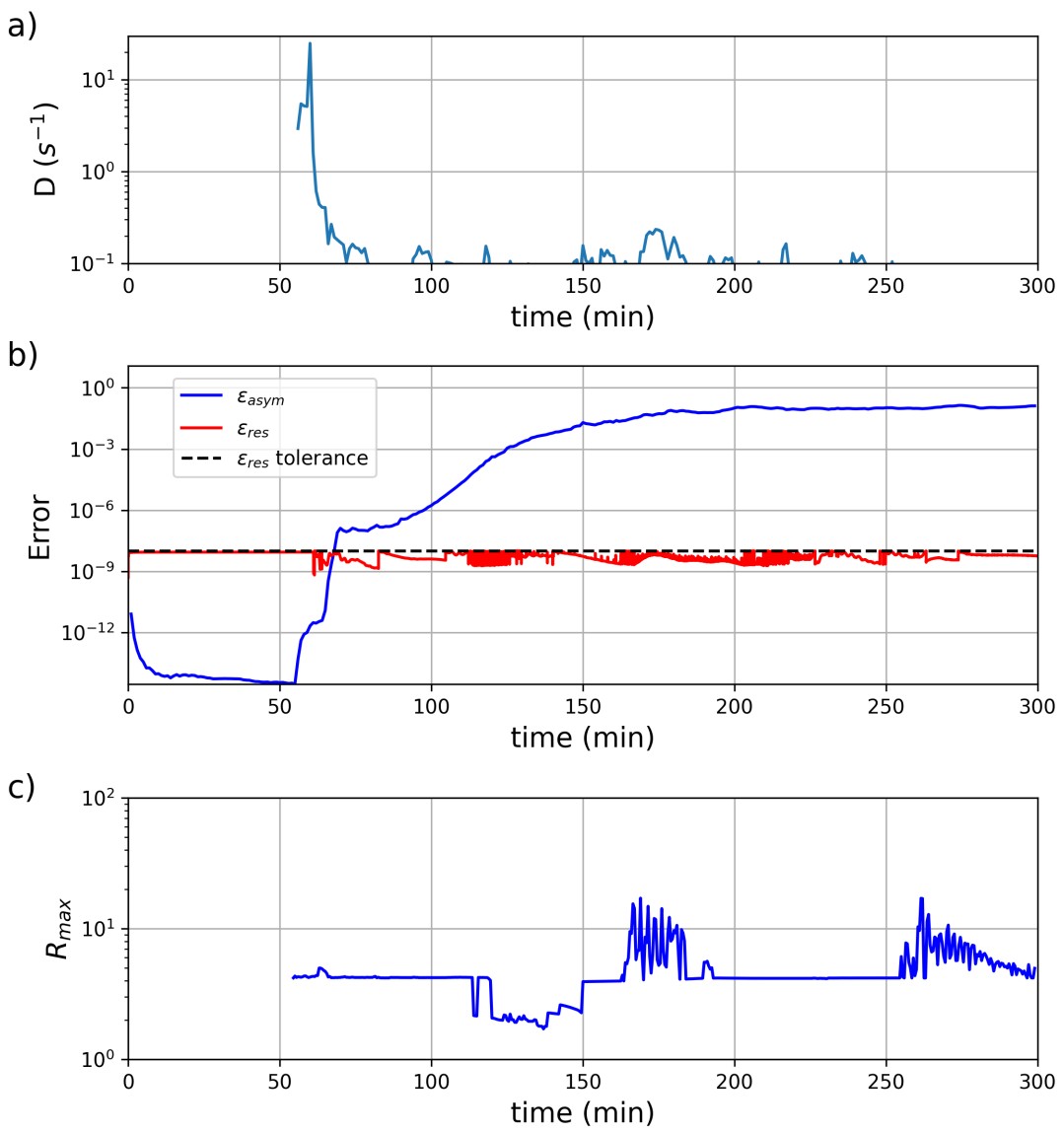

**Figure 5.** a) Temporal evolution of the damage activity $D$, b) the solution residual $\epsilon_{res}$, asymmetry factor $\epsilon_{asym}$ and convergence criterion on $\epsilon_{res}$, and c) the maximum error amplification ratio $R_{max}$, in the control simulation using the standard stress correction scheme.

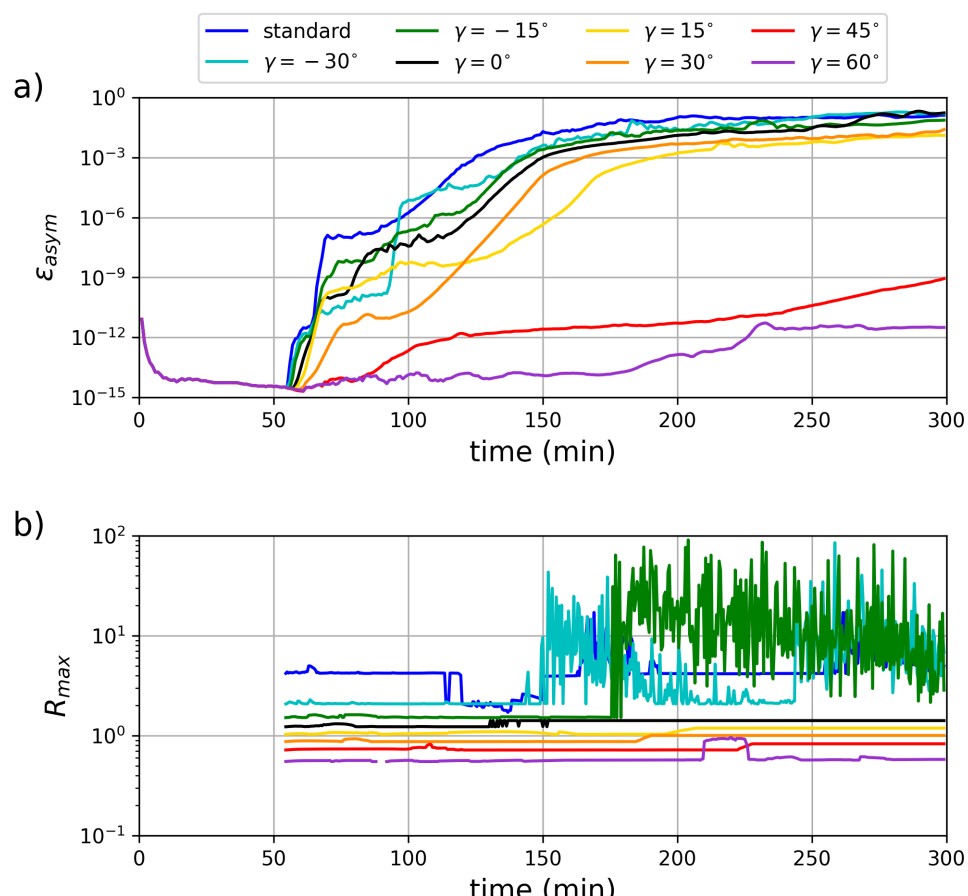

**Figure 6.** a) Time evolution of the asymmetry factor $\epsilon_{asym}$ and b) time series of the maximum error amplification ratio $R_{max}$, in a sensitivity experiment on the stress correction path angle $\gamma$, using the generalized stress correction scheme.

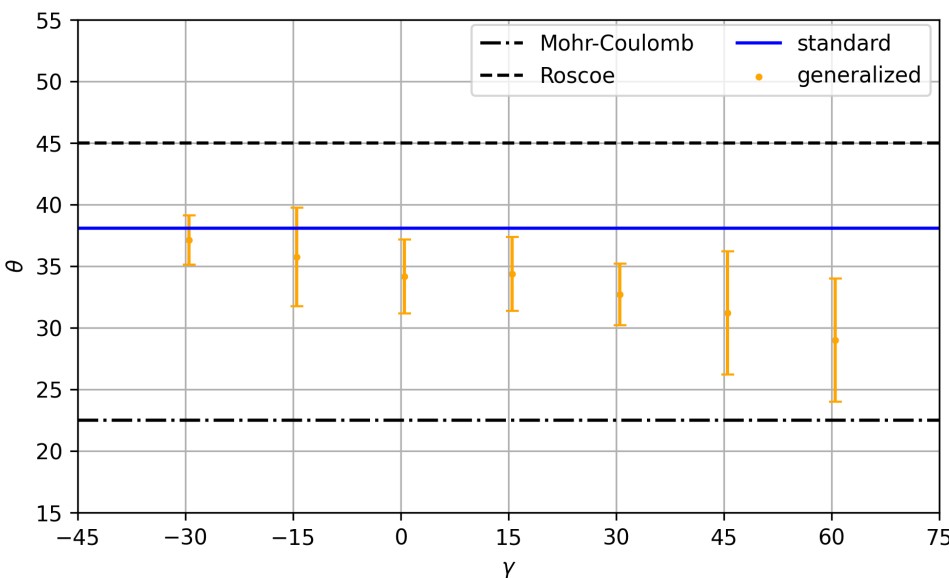

**Figure 7.** Sensitivity of the LKF orientation $\theta$ on the stress correction path angle $\gamma$ (degrees) in uniaxial loading experiments using the generalized stress correction schemes. The theoretical LKF angles from the Mohr-Coulomb and Roscoe theories are indicated by dash-dotted and dashed lines respectively for reference.

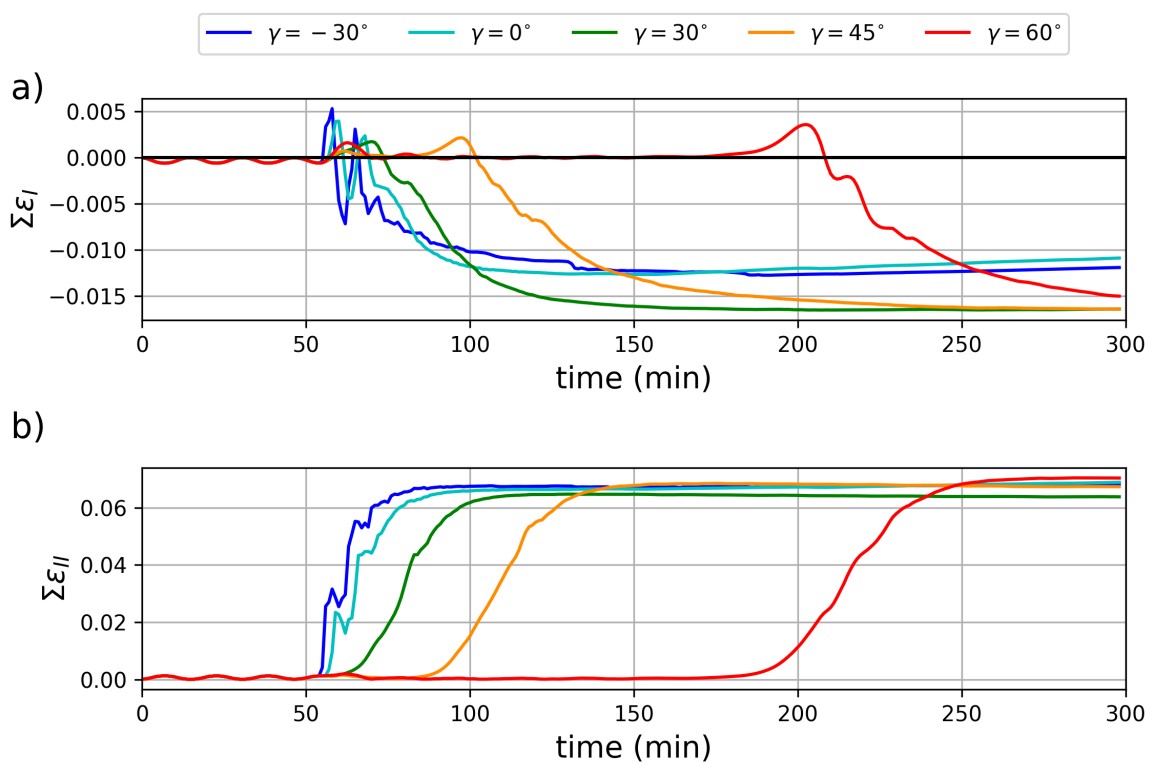

**Figure 8.** Time evolution of the mean normal (a) and maximum shear (b) strain rate invariants integrated over the ice cover, in simulations using the generalized damage parameterization with different stress correction path $\gamma$.

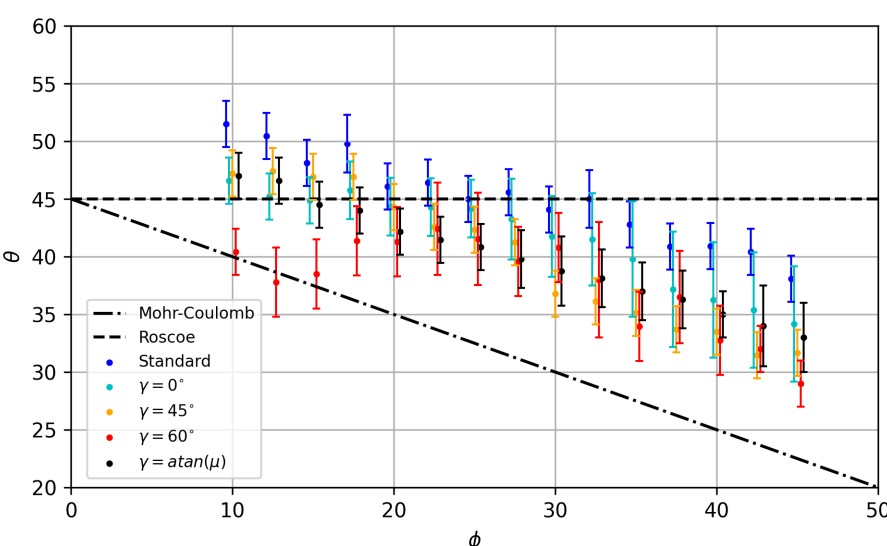

**Figure 9.** Sensitivity of the LKF orientation ($\theta$, degrees) on the angle of internal friction ($\phi$, degrees), in uniaxial loading experiments using different correction path angle ($\gamma$). The correction path angle $\gamma = atan(\mu)$ implies that the stress correction path is perpendicular to the yield curve. The theoretical LKF orientation from the Mohr-Coulomb and Roscoe theories are indicated by dash-dotted and dashed lines respectivelty for reference.

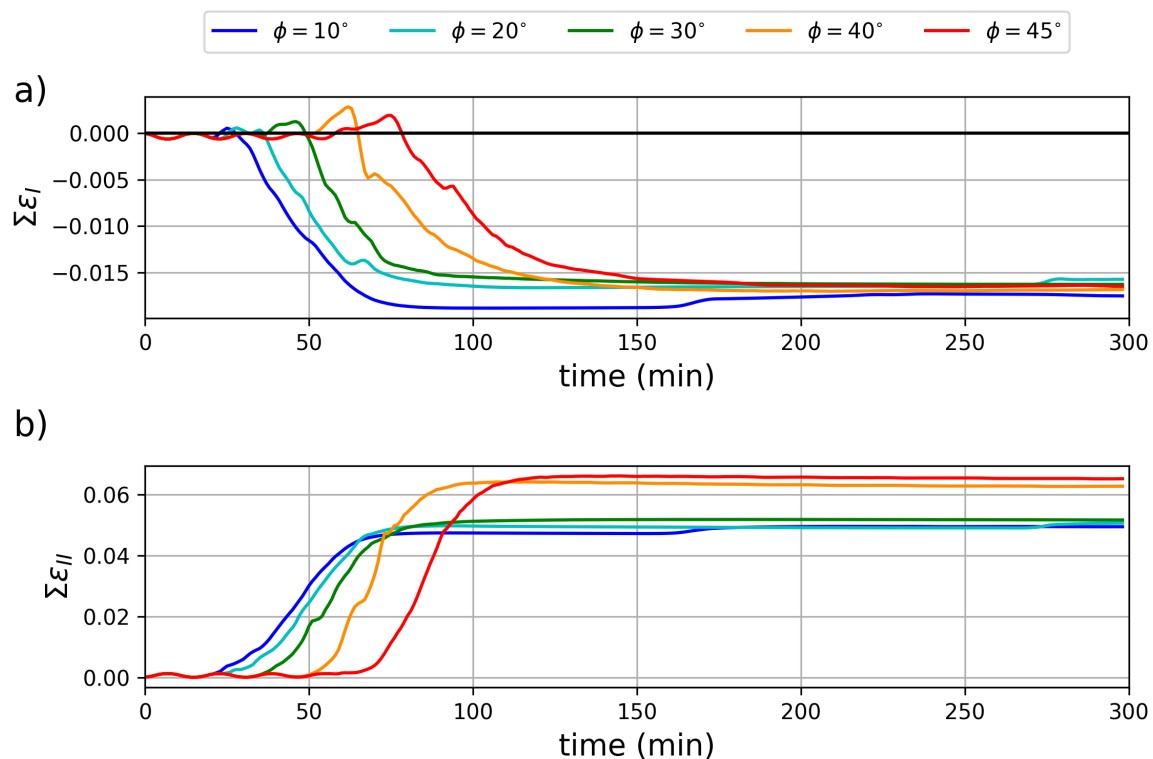

**Figure 10.** Time evolution of a) the mean normal strain rate invariant integrated over the ice cover $(\text{day}^{-1})$ and b) the maximum shear strain rate invariant integrated over the ice cover $(\text{day}^{-1})$, when using different angles of internal friction $\phi$, with a stress correction path normal to the yield curve $(\gamma = \arctan(\mu))$.

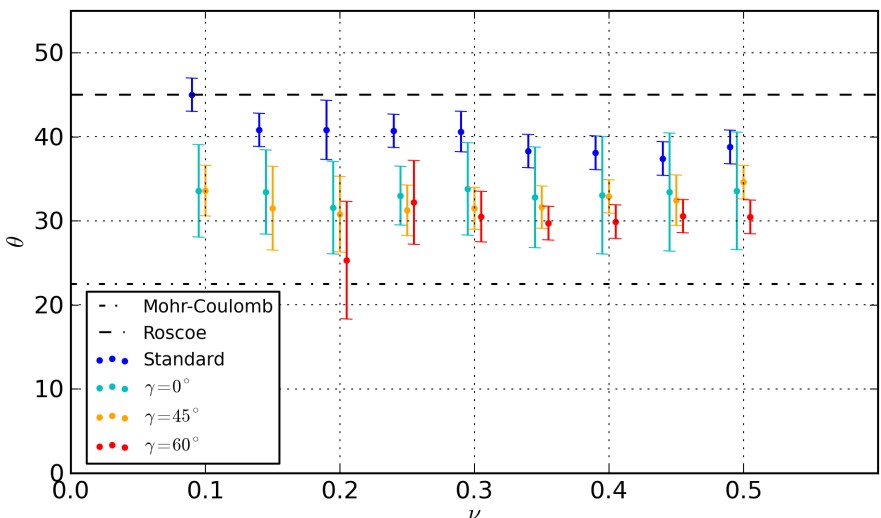

**Figure 11.** Sensitivity of the LKF orientation ($\theta$, degrees) on the Poisson ratio ($\nu$, unitless), in uniaxial loading experiments using different correction path angle ($\gamma$). The theoretical orientations from the Mohr-Coulomb and Roscoe theories are indicated by dash-dotted and dashed lines respectively for reference.

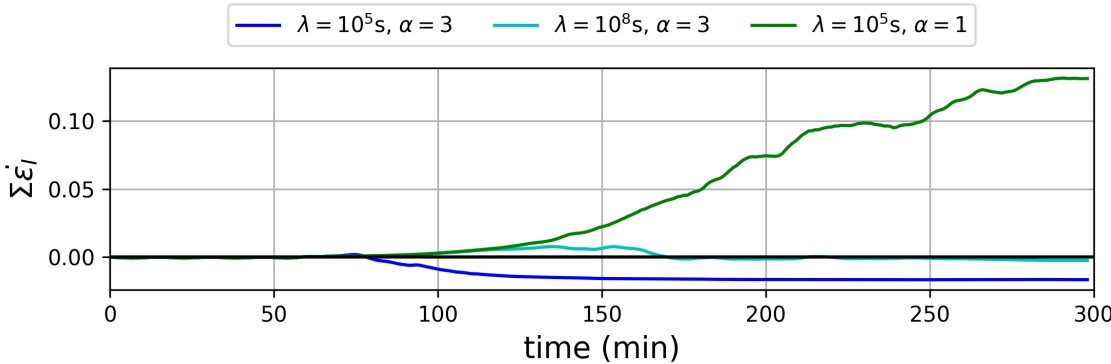

**Figure 12.** Time evolution of the mean normal strain rate invariant integrated over the ice cover (day$^{-1}$) using a stress correction path normal to the yield curve ($\gamma = \arctan(\mu)$) with $\alpha = 3$ (blue), $\alpha = 1$, and a longer viscous dissipation time-scale ($\lambda = 10^8$ s).

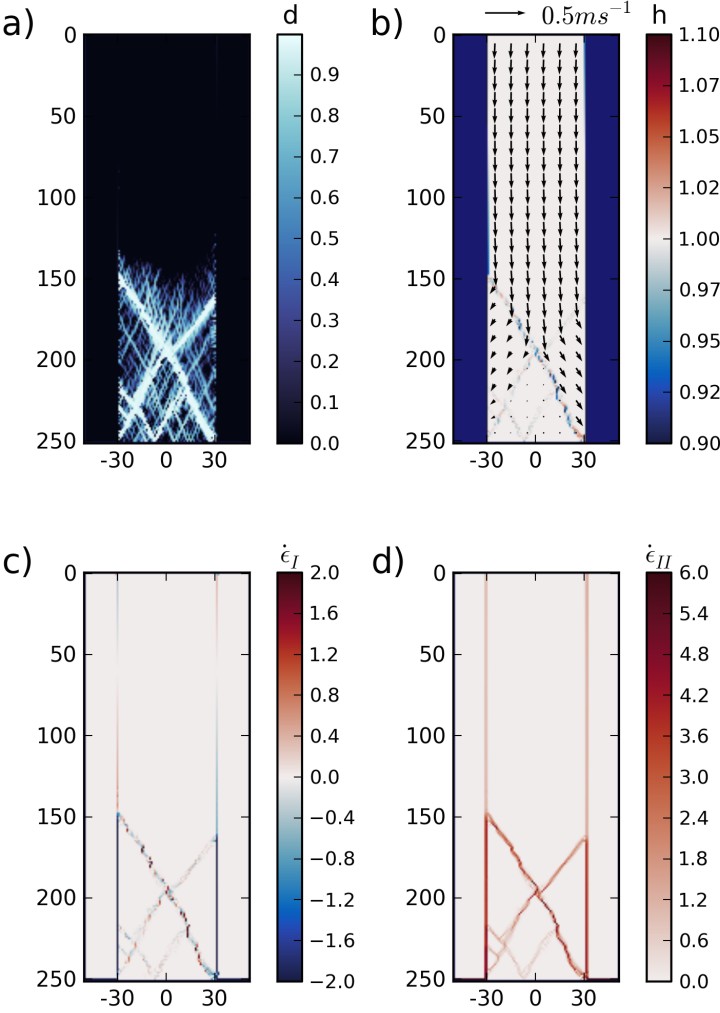

**Figure 13.** a) Damage (unitless), b) ice thickness (m, color) and velocity vectors (m s$^{-1}$), c) mean normal strain rate invariant ($\dot{\epsilon}_I$, day$^{-1}$) and d) maximum shear strain rate invariant ($\dot{\epsilon}_{II}$, days$^{-1}$) after two hours of integration in using the generalized stress correction scheme with $\gamma = 45°$ and including heterogeneity in the initial material cohesion field. The heterogeneous cohesion ($c_0$) field is defined locally at each grid cell by picking a random number between 7.0 and 13.0 kN m$^{-2}$. The remaining initial conditions are the same as all other simulations.