# Peer review of "A generalized stress correction scheme for the MEB rheology: impact on the fracture angles and deformations"

_The Cryosphere, 2020_

## Referee Comment (RC2)

**Referee's Report**
*A generalized stress correction scheme for the MEB rheology: impact on the*
*fracture angles and deformations*
by Mathieu Plante and L. Bruno Tremblay

This manuscript describes a modification of the return algorithm for supercritical stresses in the Maxwell-Elasto-Brittle (MEB) model for sea ice. The stated purpose of this modification is to better match simulated and observed fracture angles, and to reduce numerical growth of errors over the course of the simulation. The modification is tested on uniaxial deformation of a rectangular patch of sea ice. The modifications provide improvement over the previous approach but do not yet quite match observation. Overall, the goal and methods are clearly stated, although some notation is sloppy.

Trying to adjust the return algorithm to influence the failure angle is rather an indirect course of action. There does not appear to be a direct prediction of failure angle, just a demonstration through a full numerical simulation. It would be good to emphasize/clarify this point in the text (if it is in fact true), or explain how to predict the failure angle (if it is not true).

One aspect that is lacking in the presentation is the behavior of the numerical algorithm when the mesh size is changed. Fracture models are notorious for ill-posedness and it would be good to illustrate that this model's predictions do not depend on the mesh size. It is also common for the failure angle to depend on the mesh aspect ratio. Both mesh refinement and aspect ratio need to be explored.

I have some additional questions, comments, and suggested improvements.

1. Abstract: The VP model does not include fracture.

2. Page 3: You are using an Eulerian grid but I don't see equations that show advection of parameters (eg damage parameters). Are you assuming small deformations only? (See equation 12, for example.)

3. Page 4: Equation 5. Is a superposed dot the same as a partial time derivative or a material time derivative? Is this rate equation objective? Is ice deformation really rate dependent? Are there experiments about that?

4. Page 4: Equations 6 and 7: You are using multiple notations for the same thing: $x$ and $y$ components, 1 and 2 components. In Eq. 5, $\mathbf{C}$ is a fourth order tenor, Eq. 6 is a $2 \times 2$ matrix (components of a second order tensor?).

5. Page 5: Probably helpful to define $\sigma_I$ and $\sigma_{II}$ in terms of stress components.

6. Page 6: Line 150: What is the 'standard' path?

7. Page 6: Line 151: Change 'to for' to 'for'.

8. Page 6: Line 162: Schreyer et al do not use 'granular theory', assuming that means models of granular flow. It is also confusing to refer to $\sigma_c$ as a decohesive stress tensor since it has no apparent connection to Schreyer et al.

9. Page 7: Line 177: Change 'correspond' to 'corresponding'.

10. Page 7: last line: What is included in the 'solution vector'?

11. Page 8: Line 204: $\tau_a$ is a vector. I assume the scalar value you assign to it is for one component and the other is zero. (Also Page 9, Line 245.)

12. Page 9: First line: Please give a reference showing the connection between failure in granular material and sea ice under uniaxial compression.

13. Page 9: Line 226: I don't see a definition of $\delta$.

14. Page 9: Line 228: 'In general, the fracture angle ...' is this the fracture angle for sea ice?

15. Page 9: Line 244: Change 'waves' to 'wave'.

16. Page 9: Line 248: '4 cfi)' means Figure 4?

17. Page 10: Line 254: Change 'are' to 'is'. I do not see in Fig. 5 that large values of $R$ is associated with growth in $\varepsilon_{sym}$. Can you illustrate this better?

18. Page 10: Line 255: Change 'growths $R$' to 'growth in $R$'.

19. Page 10: Line 258: Change 'indicate' to 'indicates'.

20. Page 10: Line 269: Change 'depends on corrected' to 'depend on the corrected'.

21. Page 10: Lines 274-278: MEB and VP (and granular material models) make different predictions. Is there any evidence for your model behavior in experiments? The VP model is based on plasticity there is no fracture, so no 'post-fracture behavior.'

22. Page 11: Line 284: Change 'approaches' to 'approach'.

23. Page 11: Line 286: Change 'sensitive other' to 'sensitive to other'.

24. Page 11: Line 290: Change 'increase' to 'increases'.

25. Page 12: Line 332: Change 'divergence' to 'divergent'.

26. Page 12: Line 335: Change 'reach' to 'reaches'. Change 'wave' to 'waves'.

27. Page 13: Line 357: Change 'generalizes' to 'generalized'.

---

## Author Comment (AC3)

**Answers to tc-2020-354 RC2**

June 21st, 2020

Note :

- The referee comments are shown in black,
- The authors answers are shown in blue,
- *Quoted texts from the revised manuscript are shown in italic and in dark blue.*

- Note that the exact pages and line numbers in our responses are subjected to change as the revised manuscript is being prepared.

Referee's Report on: A generalized stress correction scheme for the MEB rheology: impact on the fracture angles and deformations by Mathieu Plante and L. Bruno Tremblay

This manuscript describes a modification of the return algorithm for supercritical stresses in the Maxwell-Elasto-Brittle (MEB) model for sea ice. The stated purpose of this modification is to better match simulated and observed fracture angles, and to reduce numerical growth of errors over the course of the simulation. The modification is tested on uniaxial deformation of a rectangular patch of sea ice. The modifications provide improvement over the previous approach but do not yet quite match observation. Overall, the goal and methods are clearly stated, although some notation is sloppy.

>> We thank the referee for his or her thorough review of the manuscript and constructive comments.

Trying to adjust the return algorithm to influence the failure angle is rather an indirect course of action. There does not appear to be a direct prediction of failure angle, just a demonstration through a full numerical simulation. It would be good to emphasize/clarify this point in the text (if it is in fact true), or explain how to predict the failure angle (if it is not true).

>> It is correct that we do not use the return algorithm to prescribe the fracture angle. The original goal of the study was to reduce the integration errors in the MEB rheology and study the sensitivity of the model to the return algorithm, given that the exact path along which the super-critical stresses should be returned to the yield curve is not known a priori. The fact that the fracture angle is in better agreement with observations when we use an algorithm that minimizes the error growth is a by-product of this original goal. This is clarified in the abstract at L3-5, in the introduction at L60-65 and in the discussion at L311-312 in the revised manuscript.

A more direct approach to prescribe the fracture angle would be to introduce a decohesive strain when damage increases, in the manner similar to the fracture algorithm of Schreyer et al. (2006), or Sulski and Peterson (2011). This was not included in the present parameterisation, as it would represent a significant modification of the MEB rheology, but will be considered for future model developments. These precisions added in a paragraph that we add at the end of section 2.4 in the revised manuscript.

One aspect that is lacking in the presentation is the behavior of the numerical algorithm when the mesh size is changed. Fracture models are notorious for illposedness and it would be good to illustrate that this model's predictions do not depend on the mesh size. It is also common for the failure angle to depend on the mesh aspect ratio. Both mesh refinement and aspect ratio need to be explored.

>> A more complete study of the sensitivity of the model to spatial resolution is the subject of another paper in preparation (to be submitted in the Fall). Preliminary results using a simple shear flow in a 1D channel show that the "boundary layer", or spatial scale *l* where damage occurs decreases when the spatial resolution is increased, while the number of grid points required to resolve the "boundary layer" increases. This opens the door to a series of new questions that we prefer to keep in a separate paper.

In the context of this study, we did simple uniaxial loading experiments with different spatial resolution and sample aspect ratio. We find that the simulated angle of fracture, the growth of numerical errors and the dependency of the fracture angle on the correction path are robust to the exact choice of model resolution and to the ice sample aspect ratio. This is now included in the discussion in the revised manuscript by broadening the scope of its last paragraph, previously dedicated to heterogeneity (L339-344).

We did not test the sensitivity of the results to the mesh aspect ratio. This would require significant modifications to the McGill Sea Ice Model. The code is also written using a Cartesian coordinate system and it is customary to keep dx equal to dy in such models.

 I have some additional questions, comments, and suggested improvements.

1. Abstract: The VP model does not include fracture.

>> We rephrased the problematic sentence in the abstract from: "*The post-fracture deformations are shown to be dissociated from the fracture process itself, an important difference with classical Viscous Plastic (VP) models.*", to:

 "*The post-fracture viscous deformations in the MEB model are shown to be dissociated from the fracture process itself, an important difference with classical Viscous Plastic (VP) models where plastic deformations are uniquely defined from the state of stress on the fracture plane and a flow rule*".

We also note that while the VP models do not resolve brittle fractures and the LKFs are not pre-conditioned by discontinuities in material properties, they do represent ductile fractures with simultaneous deformations that are determined by the yield stress as governed by plastic laws.

2. Page 3: You are using an Eulerian grid but I don't see equations that show advection of parameters (eg damage parameters). Are you assuming small deformations only? (See equation 12, for example.)

>> We do advect the ice thickness and concentration parameters, but neglect the advection of damage, given that the fracture process occurs in a timescale (seconds) much shorter that the advection timescale (hours). The advection of damage should be included in longer-term integration of the MEB model. Adding the advection of damage does not change the results and conclusions presented in this paper but it increases the localisation of the ice fractures. This results in higher damage values that in turn increases the rate of ridging. This has been clarified in section 2.4, L130-131 in the revised manuscript.

3. Page 4: Equation 5. Is a superposed dot the same as a partial time derivative or a material time derivative? Is this rate equation objective? Is ice deformation really rate dependent? Are there experiments about that?

>> The superposed dot is a partial time derivative. In the case of damaged ice, the large-scale sea ice deformations (especially ridging) are traditionally seen as "plastic" (see Coon et al. 1974), with stresses that are strain-rates independent that are rate-independence, in accord with laboratory experiments (Tuhkuri and Lensu, 2002). The post-fracture viscous deformations are not in accord with field observations and rather represent a simplification of the larger-scale plastic regime. These points are clarified at L112-L114 of the revised manuscript.

4. Page 4: Equations 6 and 7: You are using multiple notations for the same thing: x and y components, 1 and 2 components. In Eq. 5, C is a fourth order tenor, Eq. 6 is a 2 × 2 matrix (components of a second order tensor?).

>> We changed the indices "1" and "2" for "x" and "y" in Eq. 7, in the revised manuscript, as suggested by the reviewer.

As the reviewer points out, the tensors defined in Eq. 5 are presented in Eq. 6 and 7 using a matrix notation. This notation is often used in the literature to concisely write the components of the elastic tensor, based on the symmetry of the stress and strain tensors. It is obtained by laying out the 3 (in 2D) independent components of the 2$^{nd}$ order stress and strain tensors into a single row, such that the components of the 4$^{th}$ order tensor C are written in a 3x3 matrix (6x6 in 3D). See Rice (2010) for reference. This is clarified at L96-97 of the revised manuscript.

5. Page 5: Probably helpful to define σI and σII in terms of stress components.

>> This is added after Eq. 10 in the revised manuscript, as suggested by the reviewer.

6. Page 6: Line 150: What is the 'standard' path?

>> The "standard" path refers to the original return algorithm in the EB and MEB damage parameterization (Rampal et al. 2016, Dansereau et al. 2016), where the super-critical stresses are relaxed along a line that runs through the origin. This is now specified in a new paragraph added at the end of section 2.4 in the revised manuscript.

7. Page 6: Line 151: Change 'to for' to 'for'.

>> Corrected as suggested by the reviewer.

8. Page 6: Line 162: Schreyer et al do not use 'granular theory', assuming that means models of granular flow. It is also confusing to refer to σc as a decohesive stress tensor since it has no apparent connection to Schreyer et al.

>> We remove "granular theory" in this sentence and use the term more carefully throughout the revised manuscript.

Although our work is inspired by Schreyer et al. 2006, the name "decohesive stress tensor " is not a direct reference to their algorithm, but rather a reference to the fact that this stress is produced in association with the development of damage, hence to the decohesion of the ice material. We nonetheless note that our mention of Schreyer et al. 2006 refers to their use of a decohesive strain that is subtracted from the local elastic strain in the stress-strain relationship when the ice fractures, effectively relaxing the stress rates. This was clarified in the revised manuscript at L158-160, which now reads:

*"Note that the decohesive stress tensor used in this parameterization has a similar role as the decohesive strain used in the Elastic-Decohesive (ED) model (Schreyer et al., 2006). In the ED model, a decohesive strain represents the displacement field discontinuity in a sample associated with its cracking and relaxes the effective stress rates in the constitutive equation. It is derived from a decohesion function and depends on the mode of failure. Here, we do define a strain discontinuity associated with cracking but define the decohesive stress tensor to relax the stress states back onto the yield curve at different angles in the stress invariant space."*

9. Page 7: Line 177: Change 'correspond' to 'corresponding'.

>> Corrected as suggested by the reviewer.

10. Page 7: last line: What is included in the 'solution vector'?

>> The discretized set of equation corresponds to a system of N non-linear equations in the form of :

**A x = b ,**

where **x** is a vector formed by stacking all the $u_{i,j}$ components followed by the $v_{i,j}$ components, A is a NxN matrix with components that contains coefficients for the $u_{i,j}$ and $v_{i,j}$ dependent terms and B is a vector of length N containing the other terms. Then, the solution vector **F** is written as:

**F = A x – B .**

We chose not to include these details in the revised manuscript, as it would necessitate a lengthy numerical description, and added instead a reference to Lemieux et al., (2014) for readers in search of these precisions.

11. Page 8: Line 204: τa is a vector. I assume the scalar value you assign to it is for one component and the other is zero. (Also Page 9, Line 245.)

>> Yes. This precision is added at L204 and L245 in the revised manuscript.

12. Page 9: First line: Please give a reference showing the connection between failure in granular material and sea ice under uniaxial compression.

>> This section (4.3.3) and section 4.2 are re-written in the revised manuscript to clarify the granular character of sea ice and how it is related to fracture angles in uniaxial compression tests. A few references are added, such as Bardet et al. (1991) for granular geo-materials in uniaxial compression, Wachter et al. (2008) for fracture angles in ice samples, Wang et al. (2020) for uniaxial loading of sea ice in laboratory, Overland et al (1998) for shear bands observations in the Arctic.

13. Page 9: Line 226: I don't see a definition of $\delta$.

>> It is defined above Eq. 29 at L226, as the angle of dilatancy.

14. Page 9: Line 228: 'In general, the fracture angle ...' is this the fracture angle for sea ice?

>> We changed "In general" for "In most materials"

15. Page 9: Line 244: Change 'waves' to 'wave'.

>> Corrected as suggested by the reviewer.

16. Page 9: Line 248: '4 cfi)' means Figure 4?

>> This was an error in the labeling, and is corrected in the revised manuscript

17. Page 10: Line 254: Change 'are' to 'is'. I do not see in Fig. 5 that large values of R is associated with growth in $\varepsilon$sym. Can you illustrate this better?

>> We do not expect a correlation between R, the largest local error in the damage factor Psi, and $\varepsilon_{sym}$, which represents the domain-integrated asymmetries in the stress field. Specifically, $\varepsilon_{sym}$ corresponds to the cumulated the far-field response to all residual errors of previous time iterates. It grows with the onset of fracture (with large R values), given that the damage parameterization is the largest source of errors. With time, however, the far-field response to previous errors become large and dominates over the new errors in the damage factor Psi. We clarified this point and these variables in section 4.3.1 and at L254-L259 in the revised manuscript.

18. Page 10: Line 255: Change 'growths R' to 'growth in R'.

>> For consistency, we changed for "error amplification ratio R", as used throughout the manuscript.

19. Page 10: Line 258: Change 'indicate' to 'indicates'.

>> Corrected as suggested by the reviewer.

20. Page 10: Line 269: Change 'depends on corrected' to 'depend on the corrected'.

>> Corrected as suggested by the reviewer.

21. Page 10: Lines 274-278: MEB and VP (and granular material models) make different predictions. Is there any evidence for your model behavior in experiments? The VP model is based on plasticity there is no fracture, so no 'post-fracture behavior.'

>> As discussed in comment #3 above, the large-scale sea-ice deformations are "plastic" (see Coon et al., 1974, Tuhkuri and Lensu, 2002). The post-fracture viscous deformations in the MEB model thus do not correspond to field observations and represent a simplification of the plastic regime. The VP model simulates the plastic deformations associated with the ductile fractures, which corresponds to the observed material behaviour at the macro-scale. The VP model however does not represent the brittle component of the fractures or discontinuities in material properties, which occur at the smaller scales but may influence the fractures orientation and other deformation statistics. These points are clarified at L276-L278 in the revised manuscript.

22. Page 11: Line 284: Change 'approaches' to 'approach'.

>> Corrected as suggested by the reviewer.

23. Page 11: Line 286: Change 'sensitive other' to 'sensitive to other'.

>> Corrected as suggested by the reviewer.

24. Page 11: Line 290: Change 'increase' to 'increases'.

>> Corrected as suggested by the reviewer.

25. Page 12: Line 332: Change 'divergence' to 'divergent'.

>> Corrected as suggested by the reviewer.

26. Page 12: Line 335: Change 'reach' to 'reaches'. Change 'wave' to 'waves'.

>> Corrected as suggested by the reviewer.

27. Page 13: Line 357: Change 'generalizes' to 'generalized'.

>> Corrected as suggested by the reviewer.

References:

Bardet, J.: Orientation of shear bands in frictional soils, Journal of Engineering Mechanics - ASCE, 117, 1466–1484, https://doi.org/10.1061/(ASCE)0733-9399(1991)117:7(1466), 1991.

Coon, M. D., G. A. Maykut, R. S. Pritchard, D. A. Rothrock, and A. S. Thorndike. 1974. Modeling the pack ice as an elastic-plastic material. AIDJEX Bulletin, 24, H05.

Dansereau, V., Weiss, J., Saramito, P., and Lattes, P.: A Maxwell elasto-brittle rheology for sea ice modelling, The Cryosphere, 10, 1339–1359, https://doi.org/10.5194/tc-10-1339-2016, 2016.

Lemieux, J.-f., Knoll, D. A., Losch, M., and Girard, C.: A second-order accurate in time IMplicit – EXplicit ( IMEX ) integration scheme for sea ice dynamics, Journal of Computational Physics, 263, 375–392, https://doi.org/10.1016/j.jcp.2014.01.010, 2014.

Overland, J. E., McNutt, S. L., Salo, S., Groves, J., and Li, S.: Arctic sea ice as a granular plastic, J. Geophys. Res., 103, 21845–21868, 1998.

Rice, J. R,: Solid Mechanics, Harvard University, 2010.

Schreyer, H. L., Sulsky, D. L., Munday, L. B., Coon, M. D., and Kwok, R.: Elastic-decohesive constitutive model for sea ice, Journal of Geophysical Research: Oceans, 111, C11S26, https://doi.org/10.1029/2005JC003334, 2006.

Sulsky, D. and Peterson, K.: Toward a new elastic – decohesive model of Arctic sea ice, Physica D Nonlinear Phenomena, 240, 1674–1683, https://doi.org/10.1016/j.physd.2011.07.005, 2011.

Tuhkuri, J., and Lensu, M., Laboratory tests on ridging and rafting of ice sheets, *J. Geophys. Res.*, 107( C9), 3125, doi:10.1029/2001JC000848, 2002.

Wachter, L. M., Renshaw, C. E., & Schulson, E. M. (2009). Transition in brittle failure mode in ice under low confinement. *Acta Materialia*, **57**(2), 345-355.

Wang, Q., Lu, P., Leppäranta, M., Cheng, B., Zhang, G., & Li, Z. (2020). Physical properties of summer sea ice in the Pacific sector of the Arctic during 2008–2018. *Journal of Geophysical Research: Oceans*, 125, e2020JC016371. https://doi.org/10.1029/2020JC016371

---

## Author Comment (AC4)

**Answers to tc-2020-354 RC3**

June 21[st], 2020

Note :

- The referee comments are shown in black,
- The authors answers are shown in blue,
- *Quoted texts from the revised manuscript are shown in italic and in dark blue.*

- Note that the exact pages and line numbers in our responses are subjected to change as the revised manuscript is being prepared.

In this paper the authors introduce a modification of the MEB rheology in the form of a generalized damage parameterisation. They then proceed to test this new parameterisation using an idealised uniaxial loading setup. They find that the new parameterisation influences the resulting fracture angle, bringing it in the range of observations. The paper is well written and clear, using good English and sentence structure, and a logical flow from section to section and paragraph to paragraph.

The introduction of a modification of the MEB rheology is a niche topic, but potentially an important one and certainly one relevant for publication in the Cryosphere. As it stands, the paper has some faults I would like the authors to address. I expect they can do this adequately and that the resulting work will be fit for publication in the Cryosphere.

We thank the referee for his or her thorough review of the manuscript and constructive comments.

Major comments:

It is not clear why the authors are proposing this addition to the MEB. Is it numerics or physics, or something else? You say something general at the start, but it's vague and really only says what your modification does, not why you want to do it in the first place. This point should be crystal clear and guide the entire paper. Ideally the authors should say something like: "we want to introduce this scheme because we know it represents better the physics (and is incidentally better for the numerics). We see this by looking at the fracture angles (or some other measure)". Such a statement at the top would make this paper very strong. An admittedly overly harsh evaluation of the current state is that the authors change something for dubious reasons and get a different response – so why should we care? Is this the right result, but for the wrong reasons? I don't think that's a fair assessment, but unless the motivation is clearer it will be the impression a critical reader gets.

>> We re-wrote the introduction to better state our objectives. The goal of the study is to reduce the integration errors in the MEB rheology and study the sensitivity of the model to the stress correction scheme, given that the exact path along which the super-critical stresses should be returned to the yield

curve is not known a priori. This is clarified in the abstract, at L60-L64 and at L136-L137 in the revised manuscript.

We also add that our assessment of the sea ice deformations resulting from the use of a damage parameterization in the MEB model contributes to the current effort to assess the difference between different rheologies in reproducing satellite-derived sea-ice deformations (the FAMOS Sea-Ice Rheology Experiment (SIREx), https://epic.awi.de/id/eprint/48616/, with two papers currently under review in JGR). This is mentioned in the revised introduction.

Related to this lack of clear focus, I find it difficult to understand why you do the experiments that you do, so reading sections 4 and 5 is more demanding of the reader than it need be.

>> There is a clear need to standardized simple idealized experiments to test/evaluate different rheological models. This was identified at the workshop "Defining a cutting-edge future for sea-ice modelling" (Laugarvatn, Iceland, 2019) and again recently at the online workshop Modeling the Granular Nature of Sea Ice (https://seaicemuri.org/workshop.html). In both workshops, the simple uni-axial loading test (used in Ringeisen et al. (2019, 2020) received good acceptance for the community. This is now clarified in the introduction of the revised paper.

I also question the fact that the authors don't introduce heterogeneity into their model. They even point out themselves that it "is responsible for much of the brittle material behaviour in progressive damage models" and indicate that the residual errors are not important in a heterogeneous field – which is what MEB is supposed to give. This choice needs to be much better justified than is currently done.

>> We did not include heterogeneity in order to clearly identify the model performance (both numerics and physics). The issues related to the error growth leading to asymmetry in a problem with full symmetry and their impact on the fracture angles could not be addressed using heterogeneity. This was clarified in the revised manuscript at L339-344. We also clarify at L339-341 in the revised manuscript that the heterogeneity is responsible for the localisation and intermittency of sea ice, properties that are not investigated in our manuscript.

Finally, there's almost a hostile tone towards the MEB rheology in the discussion and conclusion section. The authors are practically gleeful in pointing out various faults of the model that are not relevant to the modifications they propose. It is of course fine to point out the faults of MEB - which apparently are plentiful – but the way it is done here borders on un-professional, in my opinion.

>> This is a serious accusation (unprofessionalism). We would ask that the reviewer identify the offending sentences and we will respond promptly whether the paper is accepted or not.

Clearly, this is not our point of view. We disagree that our tone is hostile towards the MEB model. We developed the only (to our knowledge) implementation of this rheology in a finite difference framework in order to be able to study the difference in physics independent of the numerics (other MEB implementations are done in Finite Element). Our study of the numerical and mechanical behaviour of this rheology is in the prospect of better understanding how the damage parameter simulates the deformations and to identify the key elements that can be useful to other models as we aim for higher resolution products. The current paper is a follow-up to an earlier paper where those issues were raised but not addressed. Our goal is to improve sea ice modeling in general and we believe that a multi-model approach towards this goal is very useful.

Minor comments:

L16: The formulation makes it sound as if leads and LKFs are interchangeable, but they are not.

>> We agree and removed to mention to LKF in this sentence.

L120: Shouldn't the cohesion be a function of resolution (see Weiss, 2007)? If that's the case, how do you get the same value from large scale and the lab?

>> The material strength is a function of the resolution, and we do expect smaller values at the large scales (kms in our model) than in laboratory experiments, which usually find strengths that are one or more orders of magnitude larger than what we use in our study (10 kN m$^{-2}$). Our choice of cohesion is based on results from the ice bridge experiments of Plante et al. 2020, and coherent with what has been used in other studies using the MEB rheology (e.g. Dansereau et al. 2016, 2017, 2019, and also in Rampal et al. 2016, 2019).

L135: What's the physical justification for proposing this generalised stress correction?

>> As we mentioned above, we develop the generalised stress correction in part to improve the issues identified in our previous paper, and in part to assess the influence of the super-critical stress correction on the simulated fractures and deformation, with minimal chances to the damage parameterization. This is clarified in the revised manuscript at L60-64 and at the beginning of section 3.

Note that in the original parameterization, the choice of defining the damage parameter in terms of the amount of stress in excess of the yield curve was made to offer numerical robustness and simplicity. In a perfect model for instance, this overshoot would approach zero. A physically meaningful definition of the damage parameter could involve thermodynamics relations as the stress state approaches the yield curve (see for instance Murakami 2012), or use discrete cycling methods (as in the models of Main., 2000, Amitrano and Helmstetter., 2006, Carrier et al., 2015), but would represent a significant modification of the damage parameterization. This is considered for future model development but out of the scope of this paper.

L222: Mohr-Coulomb and Roscoe theories both concern granular materials, but hear we're dealing with the fracturing of a solid. Are they still valid? Please elaborate.

>> Sea ice is a granular material. Sea for instance books from Leppäranta (2011), Weiss (2013), and the recent workshop "Modeling the Granular Nature of Sea Ice" (https://seaicemuri.org/workshop.html), bringing scientists from all around the world working on this topic, for reference.

L248: There's a lot of information in figure 4 and the reader needs more help in deducing why you created it and what it's supposed to tell us.

>> Additional information is included in the figure caption of the revised manuscript, which now reads:

"Scatter plots of local stress invariants ($\sigma_{I}$ vs. $\sigma_{II}$, in kN m$^{-1}$, left column), normal stresses and scaled strain rate invariants ($\sigma_{I}$ vs. (1-d)^3 $\dot{\epsilon}_{II}$, right column) in heavily damaged (d > 0.9) grid cells, at t = 60 min (shortly after the fracture, top row), t = 120 min (~1 hour after the fracture, middle row), and t = 180 min ( ~2 hours after the fracture, bottom

*row). Color indicates the local damage. The strain rates are normalized to account for the non-linear dependency of the viscosity $\eta$ on the damage parameter. The gradual alignment of the points in the $\sigma_{I}$ vs. $(1-d)^3 \dot{\epsilon}_{II}$ diagram indicate the development of a linear-viscous stress-strain relationship over time."*

L276: A reference to the contrasting results is needed.

>> We add the reference to Ringeisen et al. (2019) for the VP model and to Bardet (1991), Balendran and Nemat-Nasser (1993), for granular materials.

L291: A reference for what is typical for granular material is needed (a textbook will suffice).

>> We added the reference to the book from J. Duran (1999), and to Bardet (1991). See references below.

L316: This entire paragraph is a bit up-side-down to me. You start by saying the MEB is not good enough, for various reasons (begging the question of why you use it in the first place, actually) - and then you say how your new addition will not save it. A more natural way to write this is to first say that although the decohesive stress tensor can do some things it cannot fix everything, including etc.

>> We do not state that the MEB is not good enough. We mention the differences in behaviour with respect to the more commonly used VP models and discuss these differences in terms of potential limitations that should be taken into account in future model developments. We do believe that the use of several different models raises questions that would not be raised with the use of a single model, even if that model is better. Our community has suffered from a monopoly in approach with the standard VP model, until only very recently when new approaches were developed. This is made clear in the introduction of the revised manuscript.

We also believe that this discussion is made clearer in the revised manuscript by specifying in the abstract, at L60-64 and L136-L137 that we developed the generalized damage parameterization in part to investigate the influence of the return algorithm on the simulated fractures.

L329: This paragraph is off topic, discussing experiments not introduced before and not relevant to the introduction of the decohesive stress tensor. Please remove.

>> We argue that this paragraph serves to put our results in context with other MEB model studies, which use different material parameters. In the revised manuscript, we widen the discussion to integrate the effect of grid resolution, sample aspect ratio, advection and heterogeneity, and clarify the context of this discussion.

L353: Now I'm confused, did you want to solve the ridging problem by introducing the decohesive stress? Again, a more natural way to present your results would be to first state what works and then what remains.

>> We now specify that the generalized damage parameterization modification is used to tackle issues that we raised on the damage parameterization in a previous paper (Plante et al. 2020) but also to investigate the influence of the return algorithm on the simulated fractures. The dominance of the post-fracture deformations in the MEB rheology is an important finding in our experiments, which contrasts

with the behaviour in the VP and EVP rheologies. This is clarified in the conclusion at L346-348, but also in the abstract, at L60-L64 and at L136-L137in the revised manuscript.

References:

Amitrano, D. and Helmstetter, A.: Brittle creep, damage and time to failure in rocks, Journal of Geophysical Research : Solid Earth, 111, B11 201, https://doi.org/10.1029/2005JB004252, 2006.

Balendran, B. and Nemat-Nasser, S.: Double sliding model for cyclic deformation of granular materials, including dilatancy effects, J. Mech. Phys. Solids, 41, 1993.

Bardet, J.: Orientation of shear bands in frictional soils, Journal of Engineering Mechanics - ASCE, 117, 1466–1484, https://doi.org/10.1061/(ASCE)0733-9399(1991)117:7(1466), 1991.

Carrier, A., Got, J.-L., Peltier, A., Ferrazzini, V., Staudacher, T., Kowalski, P., and Boissier, P.: A damage model for volcanic edifices: Implications for edifice strength, magma pressure, and eruptive processes, Journal of Geophysical Research: Solid Earth, 120, 567–583, https://doi.org/10.1002/2014JB011485, 2015.

Dansereau, V., Weiss, J., Saramito, P., and Lattes, P.: A Maxwell elasto-brittle rheology for sea ice modelling, The Cryosphere, 10, 1339–1359, https://doi.org/10.5194/tc-10-1339-2016, 2016.

Dansereau, V.,Weiss, J., Saramito, P., Lattes, P., and Coche, E.: Ice bridges and ridges in the Maxwell-EB sea ice rheology, The Cryosphere, 11, 2033–2058, 2017.

Dansereau, V., V. D_emery, E. Berthier, J. Weiss, and L. Ponson.: Collective Damage Growth Controls Fault Orientation in Quasibrittle Compressive Failure, Phys. Rev. Lett., 122, 085,501, doi:10.1103/PhysRevLett.122.085501, 2019.

Duran, J., *Sands, Powders, and Grains: An Introduction to the Physics of Granular Materials* (translated by A. Reisinger). November 1999, Springer-Verlag New York, Inc., New York.

Leppäranta M.,: *The drift of sea ice*, Springer-Verlag Berlin Heidelberg, doi:10.1007/b138386, 2011.

Main, I. G.: A damage mechanics model for power-law creep and earthquake aftershock and foreshock sequences, Geophysical Journal International, 142, 151–161, https://doi.org/10.1046/j.1365-246x.2000.00136.x, 2000.

Plante, M., Tremblay, B., Losch, M., and Lemieux, J.-F.: Landfast sea ice material properties derived from ice bridge simulations using the Maxwell elasto-brittle rheology, The Cryosphere, 14, 2137–2157, https://doi.org/10.5194/tc-14-2137-2020, https://tc.copernicus.org/articles/14/2137/2020/, 2020.

Rampal, P., Bouillon, S., Ólason, E., and Morlighem, M.: neXtSIM: a new Lagrangian sea ice model, The Cryosphere, 10, 1055–1073, https://doi.org/10.5194/tc-10-1055-2016, 2016.

Rampal, P., Dansereau, V., Olason, E., Bouillon, S., Williams, T., Korosov, A., and Samaké, A.: On the multi-fractal scaling properties of sea ice deformation, The Cryosphere, 13, 2457–2474,

https://doi.org/10.5194/tc-13-2457-2019, 2019.

Ringeisen, D., Losch, M., Tremblay, L. B., and Hutter, N.: Simulating intersection angles between conjugate faults in sea ice with different viscous–plastic rheologies, The Cryosphere, 13, 1167–1186, https://doi.org/10.5194/, 2019.

Ringeisen, D., Tremblay, 490 L. B., and Losch, M.: Non-normal flow rules affect fracture angles in sea ice viscous-plastic rheologies, The Cryosphere Discussions, 2020, 1–24, https://doi.org/10.5194/tc-2020-153, https://tc.copernicus.org/preprints/tc-2020-153/, 2020.

Weiss J. Drift, Deformation and Fracture of Sea Ice: A Perspective Across Scales. Netherlands: Springer, Dordrecht; 2013, p. 83. https://doi.org/10.1007/978-94-007-6202-2.

Weiss, J., Schulson, E. M., and Stern, H. L.: Sea ice rheology from in-situ, satellite and laboratory observations : Fracture and friction, Earth and Planetary Science Letters, 255, 1–8, https://doi.org/10.1016/j.epsl.2006.11.033, 2007.

---

## Author Response (AR1)

**Authors answers to tc-2020-354 comments**

**May 6th, 2021**

Dear Editor,

We are pleased to submit the revised manuscript of our paper entitled "A generalized stress correction scheme for the MEB rheology: impact on the fracture angles and deformations" by Mathieu Plante and L. Bruno Tremblay.

We would like to thank the reviewers for their useful comments and suggestions. We have modified our manuscript according to most suggestions of the reviewers. This helped improve the clarity of the article substantially.

Thank you for your consideration for publication.

Sincerely,

On behalf of all the authors,
Mathieu Plante

Note:

- The referee comments are shown in black,
- The authors answers are shown in blue,
- *Quoted texts from the revised manuscript are shown in italic and in dark blue.*
- Amendments made to the responses in the open discussion are shown in green

**Answers to tc-2020-354 RC1**

June 21st, 2020

Note:

- The referee comments are shown in black,
- The authors answers are shown in blue,
- *Quoted texts from the revised manuscript are shown in italic and in dark blue.*
- Amendments made to the responses in the open discussion are shown in green

Review of "A generalized stress correction scheme for the MEB rheology: impact on the fracture angles and deformations" by Plante and Tremblay (tc-2020-354).

This manuscript is a generally well written and easy to follow description of an extension to the MEB model of Plante et al (2020), but also has implications for other MEB implementations. This extension addresses two problems of the original model: large (numerical) error growth, which may be model code specific) and too large fracture angles (which is probably not model-code specific). The new scheme allows to specify a more general correction of stress states that exceed the Mohr-Coulomb failure criterion. Sensitivity experiments in uniaxial compression illustrate that the parameterization indeed reduces the error growth and also reduces the fracture angles towards more realistic values. From the sensitivity experiments a preferred parameter set is determined. This is a very useful addition to the development of MEB rheology (and code) and should be published subject to minor revisions.

We thank the referee for his or her thorough review of the manuscript and constructive comments.

My main point of critique is that the manuscript is missing a bit of general introduction and a clear problem statement. To my mind, the manuscript can be improved by taking the reader more by the hand than is done. This only requires a few sentences here and there or maybe an additional paragraph, e.g.

(1) what do we expect from a "brittle" model in contrast to a "granular material". The concept of "granular material/flow" is used often in the text, but it is not clear (from the text) how the brittle part of the model relates to that. Do we expect that a brittle model represents a granular material properly?

Brittle and ductile are types of fractures: a brittle fracture occurs with little prior plastic (permanent) deformation, and ductile fracture occurs after significant plastic deformations. A granular material, on the other hand, is a type of material: i.e., a composite of aggregated

granules, as opposed to metals or crystals). Granular theories describe the fractures and deformations of granular materials in terms of the distribution of contact normal between individual grains.

A brittle model is thus expected to represent the nucleation and propagation of cracks in a material, effectively producing a material discontinuity in the material properties. A granular model typically uses a Mohr-Coulomb yield curve and granular flow rules (sliding along fracture planes with more or less dilatation) to govern the material dynamics.

In the MEB model for instance, the damage parameterization corresponds to the "brittle" behaviour of sea ice, and the granular behaviour is reflected in the choice of a Mohr-Coulomb yield curve.

This is now clarified in a new paragraph at the beginning of the model section 2.2, L116-120 and in section 4.3.3.

(2) state the issues with sea models and MEB in particular that are addressed in this paper in separate paragraphs. Now the angle-issue is mentioned in the middle of a paragraph that is introduced by: "The damage parameterization is relatively new, …"

>> We re-wrote the introduction (as suggested) to better introduce these issues, as well as addressing the previous and following comments from the reviewer. We address the challenges in representing the fracture of sea ice in the context of large-scale sea-ice models that are based on the continuum assumption, and then address issues that are more specific about the MEB rheology.

(3) discuss if the new scheme can be also useful for other implementations of MEB (e.g. neXtSim)

>> Yes, the new scheme can be used in other implementation of the MEB model (e.g. neXtSIM). Specifically, our implementation represents a generalization of the damage parameterization that can be easily implemented numerically and used to improve the performance of MEB models. Our results also show that the new scheme can be used to tune the simulated fractures closer to observations. These statements are added in the discussion section, L398-406 and in conclusions.

There are some technical issues (figure referencing and captions) detailed below.
The points below sometimes repeat my main points.

page 1

Abstract: General background and (more importantly) a clear problem statement is missing from the abstract. E.g., the problem of too large angles is not stated and the error growth is also only mentioned as the target of the new parameterization. It would help to have more context here already (1-2 extra sentences).

>> A clearer statement for the motivation of this study in included in the revised manuscript. The

abstract now reads:

*"The Maxwell Elasto-Brittle (MEB) rheology uses a damage parameterization to represent the brittle fracture of sea ice without involving plastic laws to constrain the sea-ice deformations. The MEB damage parameterization is based on a correction of super-critical stresses that binds the simulated stress to the yield criterion but leads to a growth of errors in the stress field. A generalized damage parameterization is developed to reduce this error growth and to investigate the influence of the super-critical stress correction scheme on the simulated sea-ice fractures, deformations and orientation of Linear Kinematic Features (LKFs). A decohesive stress tensor is used to correct the super-critical stresses towards different points on the yield curve. The sensitivity of the simulated sea-ice fractures and deformations to the decohesive stress tensor is investigated in uniaxial compression experiments. Results show that the decohesive stress tensor influences the growth of residual errors associated with the correction of super-critical stresses, the orientation of the lines of fracture and the short-term deformation associated with the damage, but does not influence the long-term post-fracture sea-ice deformations. We show that when ice fractures, divergence first occurs while the elastic response is dominant, and convergence develops post-fracture in the longer-term when the viscous response dominates -- contrary to laboratory experiment of granular flow and satellite imagery in the Arctic. The post-fracture deformations are shown to be dissociated from the fracture process itself, an important difference with classical Viscous Plastic (VP) models in which large deformations are governed by associative plastic laws. Using the generalized damage parameterization together with a stress correction path normal to the yield curve reduces the growth of errors sufficiently for the production of longer-term simulations, with the added benefit of bringing the simulated LKF angles closer to observations (from $40-50^\circ$ to $35-45^\circ$, compared to $20-30^\circ$ in observations)."*

l3: any correction path: unclear "any"

>> This was replaced by "*towards different points on the yield curve*" in the revised manuscript.

l18: significantly: repetition

>> Corrected as suggested by the reviewer.

l22: the presence of and deformations along LKFs

>> Corrected as suggested by the reviewer.

l30: Hunke, 2001: not sure if this is an appropriate reference (for what)?

>> We removed this reference in the revised manuscript.

l44: "The fracture angle simulated by the MEB and standard VP models" It will be easier to follow, if you dedicate a separate paragraph (or at least an introductory sentence to this paragraph) to the fracture angles as a problem statement before describing what VP and MEB models do wrong.

>> We agree with the reviewer. We revised the introduction and added a new paragraph that focuses on the representation of fracture angles in both the VP and MEB models.

ll60 I think that the problem statement is not clear enough. Unless you are very familiar with the details of the implementation of MEB models, it's not clear where Plante et al (2020) had numerical difficulties and if this is specific to their implementation. It should be clear if this will also be of value for, e.g. neXtSIM, or Dansereau et al.

Also the fracture angle problem is somewhat buried in the introduction and should be more prominent, because the paper devotes a large part to this.

>> We now devote a paragraph on the MEB model behaviour where these points are clarified as suggested. We specify that the numerical difficulty is related to the integration of the residual errors in the damage parameter, and is associated with the damage equation used in all MEB models.

l62: (Sulsky and Peterson, 2011) fix parentheses

>> Corrected as suggested by the reviewer.

l81: (Plante et al., 2020) fix parentheses

>> Corrected as suggested by the reviewer.

l96: is -> in

>> Corrected as suggested by the reviewer.

l104: "resulting in dominant elastic component"? not clear, something missing?

>> This refers to the dominance of the elastic term vs. the negligible viscous term in the constitutive equation. We clarified these lines in the revised manuscript, which now read:

*"[...] the elastic term dominates when the ice is undamaged while the viscous term dominates when the ice is heavily fractured."*

ll117: maybe put \mu, \phi, c into Fig1 for better illustration?

>> We added the parameters as suggested by the reviewer.

eq 16: where does the "some algebra" start from? Maybe add a little more explanation here to guide the reader.

>> We added more information in the revised manuscript. Eq. 19 (in the revised manuscript) is found by finding the intersection point between the yield curve (Eq. 10) and the line

corresponding to the stress correction. We add the mathematical expression of the stress correction line, such that the algebra is more straightforward.

l148: "something that is not possible in the standard parameterization otherwise \Psi …" please rephrase.

>> This sentence is clarified in the revised manuscript, and now reads:

*"[…] as opposed to the standard parameterization in which case any super-critical stress is returned to the origin."*

l166: on -> of

>> Corrected as suggested by the reviewer.

l208: asymmetry factor: not immediately clear why this measures error. I assume that you expect perfectly symmetric solutions about the center line, but I think that this needs to be explained.

>> This diagnostic is explained in more details in section 4.3.1, L274-L285 in the revised manuscript, including its definition about the center line. We specify that it measures the cumulated far-field response to all residual errors produced from the start of the simulation. As opposed to the amplification factor R, which only measures the maximum local amplification of the residual error by the damage parameterization, the asymmetry measures the cumulative and longer-term effect of the residual errors on the model solution.

The same is true for "damage activity", what do you want to use this for and how does this diagnostic achieve that.

>> The damage activity is only used to indicate the onset of fracturing and the short time scale associated with the development of fracture. It serves to show that the onset of the growth of errors in the solution is associated with the fracture. This is specified in section 4.3.2, L290-292 in the revised manuscript.

eq.26/27. the notation is a bit unusual and looks a little like (pseudo-) code. Why not use standard indexing as one would expect in a maths text?, e.g. \left(\sigma_{II}\right)_{n_x-i,j}

>> We agree and the format is corrected in the revised manuscript.

l235: 0.29 N/m? units?

>> This is an error and is corrected to N m$^{-2}$.

l243: "mostly elastic with divergence along the fracture line" Where do we see that divergence? In Fig3 I mostly see negative divergence = convergence.

>> This is illustrated in Fig. 4b (the reference is added in the revised manuscript). Figure 3 shows the deformations after 2 hours of simulations, in which points the deformations are dominated by the post-fracture (viscous) convergence. This is also clarified at the beginning of this paragraph, at L322-324 in the revised manuscript.

l248/9 The references to figure 4 are not correct. There is no Fig 4i, then it's not clear from the caption, what we are seeing in color (damage?). It would help to add the timing in the plot (maybe top right or bottom left of rhs column).

>> There were errors in the labelling. This is corrected in the revised manuscript. We also improved the labels and captions in this figure.

l251: here and everywhere else: Units should NOT be in italics).

>> Corrected as suggested by the reviewer.

l254: $10^{-6}$ Nm$^{-2}$ (unit not in italics): in 4.1 it was 1e-8!! In Fig5 it seems to be 1e-8 as well.

>> It should indeed indicate 10-8, this is corrected in the revised manuscript.

l254:are -> is

>> Corrected as suggested by the reviewer.

l254: "damage error amplification ratio R" maybe refer to equation 23 here?

>> We agree and added the reference in the revised manuscript.

l258: indicate -> indicates

>> Corrected as suggested by the reviewer.

In Figure 6 the panels for eps_asym and R_max are exchanged wrt to Figure 5. Why confuse the reader?

>> We agree with the reviewer and interchanged the panels in the revised manuscript.

l263: "the production of" could be removed

>> Corrected as suggested by the reviewer.

l264: I would argue for \gamma \ge 0 the improvement is significant (including 0). But the asymmetry also grows for \gamma > 0 and only for values > 45 it seem to stay low. Why not discuss that here?

>> We added a few lines in the revised manuscript in section 5.2, L341-350 to address this comment, instead of only bringing this point in the discussion section. We note that the improvement by the generalized parametrization is limited by the fact that the damage remains an integrated parameter, and that the residual error remains very influential in heavily damaged ice due to by the non-linear relationship between the sea ice deformation and the damage. We specify that the main improvement here is the removal of the spikes in the amplification ratio. We also note that the slower growth of asymmetries in the case of large correction angles are partly attributed to the slower development of the discontinuity. Thus, as we increase gamma, the improvement comes increasingly at the cost of losing the brittle behaviour of sea ice.

l286: "Based on these results, we suggest the use of a correction path that is normal to the yield criterion (\gamma = arctan \mu, see black points in Fig. 9)." my say, \gamma = \phi in this case, (isn't it)?

>> It is not the case. \gamma and \mu are defined in the stress invariant space, whereas the friction angle \phi is defined in the Mohr stress space. That is, the angle of friction \phi does not correspond to the angle spanned from the x axis to the yield curve in the stress invariant space. Rather, the slope of the yield curve in the stress invariant space is \mu = sin(\phi). Thus, writing \gamma as a function of \phi would yield: \gamma = arctan( sin(\phi) ).

l331: "are robust to the exact" -> are not sensitive to the exact, are robust with respect to the exact …

>> Corrected as suggested by the reviewer.

l335: reach -> reaches

>> Corrected as suggested by the reviewer.

l335: elastic wave are however no-longer -> elastic waves, however, are no longer …

>> Corrected as suggested by the reviewer.

l344: in -> is

>> Corrected as suggested by the reviewer.

l349: the uniaxial -> a uniaxial

>> Corrected as suggested by the reviewer.

l351: not sure if "post-fracture" (or pre-fracture) is grammatically correct. I would use "after (and before) fracture" in most places in this manuscript

>> We prefer to keep "post-fracture" as it is concise and often used in the field to describe material behaviour.

l352: "contrary to laboratory experiments of granular materials and satellite observations of sea ice." A short discussion about to what extent we expect granular behavior in an MEB model seems in place (not here in the conclusions but somewhere in the introduction?), in order to understand if this is an encouraging or a discouraging result

>> We agree with the reviewer and provide more background on the fracture angles and dilatancy in section 4.3.3. This section (4.3.3) and section 4.1 in the revised manuscript are re-written to clarify the granular character of sea ice and how it is related to fracture angles in uniaxial compression tests.

l361: "the production of" remove, see above

>> Corrected as suggested by the reviewer.

Figure3: miximum -> maximum/minimum?

>> This is corrected to "maximum" in the revised manuscript.

Figure 4. What are the meaning and the units of the color scale? Is this for the control simulation only?

>> This figure is for the control run and the color indicates the local damage (unitless) of each scatter points. These precisions are added in the revised manuscript.

Fig9: "The theoretical fracture angle from the Mohr-Coulomb and Roscoe theories are indicated by dashed and dash-dotted lines for reference." something like this, could also be useful in Fig7.

>> We agree and added these precisions in the captions of both Fig 7 and 9

**Answers to tc-2020-354 RC2**

June 21[st], 2020

Note:

- The referee comments are shown in black,
- The authors answers are shown in blue,
- *Quoted texts from the revised manuscript are shown in italic and in dark blue.*
- Amendments made to the responses in the open discussion are shown in green

Referee's Report on: A generalized stress correction scheme for the MEB rheology: impact on the fracture angles and deformations by Mathieu Plante and L. Bruno Tremblay

This manuscript describes a modification of the return algorithm for supercritical stresses in the Maxwell-Elasto-Brittle (MEB) model for sea ice. The stated purpose of this modification is to better match simulated and observed fracture angles, and to reduce numerical growth of errors over the course of the simulation. The modification is tested on uniaxial deformation of a rectangular patch of sea ice. The modifications provide improvement over the previous approach but do not yet quite match observation. Overall, the goal and methods are clearly stated, although some notation is sloppy.

>> We thank the referee for his or her thorough review of the manuscript and constructive comments.

Trying to adjust the return algorithm to influence the failure angle is rather an indirect course of action. There does not appear to be a direct prediction of failure angle, just a demonstration through a full numerical simulation. It would be good to emphasize/clarify this point in the text (if it is in fact true), or explain how to predict the failure angle (if it is not true).

>> It is correct that we do not use the return algorithm to prescribe the fracture angle. The original goal of the study was to reduce the integration errors in the MEB rheology and study the sensitivity of the model to the return algorithm, given that the exact path along which the super-critical stresses should be returned to the yield curve is not known a priori. The fact that the fracture angle is in better agreement with observations when we use an algorithm that minimizes

the error growth is a by-product of this original goal. This is clarified in the abstract at L5-7, L17-18, in the introduction at L86-88 and in the discussion in the revised manuscript.

A more direct approach to prescribe the fracture angle would be to introduce a decohesive strain when damage increases, in the manner similar to the fracture algorithm of Schreyer et al. (2006), or Sulski and Peterson (2011). This was not included in the present parameterisation, as it would represent a significant modification of the MEB rheology, but will be considered for future model developments. These precisions added in a paragraph that we add at L214-219 in the revised manuscript.

One aspect that is lacking in the presentation is the behavior of the numerical algorithm when the mesh size is changed. Fracture models are notorious for illposedness and it would be good to illustrate that this model's predictions do not depend on the mesh size. It is also common for the failure angle to depend on the mesh aspect ratio. Both mesh refinement and aspect ratio need to be explored.

>> A more complete study of the sensitivity of the model to spatial resolution is the subject of another paper in preparation (to be submitted in the Fall). Preliminary results using a simple shear flow in a 1D channel show that the "boundary layer", or spatial scale $l$ where damage occurs decreases when the spatial resolution is increased, while the number of grid points required to resolve the "boundary layer" increases. This opens the door to a series of new questions that we prefer to keep in a separate paper.

In the context of this study, we did simple uniaxial loading experiments with different spatial resolution and sample aspect ratio. We find that the simulated angle of fracture, the growth of numerical errors and the dependency of the fracture angle on the correction path are robust to the exact choice of model resolution and to the ice sample aspect ratio. This is now included in the discussion in the revised manuscript by broadening the scope of its last paragraph, previously dedicated to heterogeneity (L430-441).

We did not test the sensitivity of the results to the mesh aspect ratio. This would require significant modifications to the McGill Sea Ice Model. The code is also written using a Cartesian coordinate system and it is customary to keep dx equal to dy in such models.

I have some additional questions, comments, and suggested improvements.

1. Abstract: The VP model does not include fracture.

>> We rephrased the problematic sentence in the abstract from: "*The post-fracture deformations are shown to be dissociated from the fracture process itself, an important difference with classical Viscous Plastic (VP) models.*", to:

 "*The post-fracture viscous deformations in the MEB model are shown to be dissociated from the fracture process itself, an important difference with classical Viscous Plastic (VP) models in which large deformations are governed by associative plastic laws*".

We also note that while the VP models do not resolve brittle fractures and the LKFs are not pre-conditioned by discontinuities in material properties, they do represent ductile fractures with simultaneous deformations that are determined by the yield stress as governed by plastic laws.

2. Page 3: You are using an Eulerian grid but I don't see equations that show advection of parameters (eg damage parameters). Are you assuming small deformations only? (See equation 12, for example.)

>> We do advect the ice thickness and concentration parameters, but neglect the advection of damage, given that the fracture process occurs in a timescale (seconds) much shorter that the advection timescale (hours). The advection of damage should be included in longer-term integration of the MEB model. Adding the advection of damage does not change the results and conclusions presented in this paper but it increases the localisation of the ice fractures. This results in higher damage values that in turn increases the rate of ridging. This has been clarified in section 2.4, L165-169 in the revised manuscript.

3. Page 4: Equation 5. Is a superposed dot the same as a partial time derivative or a material time derivative? Is this rate equation objective? Is ice deformation really rate dependent? Are there experiments about that?

>> The superposed dot is a partial time derivative. In the case of damaged ice, the large-scale sea ice deformations (especially ridging) are traditionally seen as "plastic" (see Coon et al. 1974), with stresses that are strain-rates independent that are rate-independence, in accord with laboratory experiments (Tuhkuri and Lensu, 2002). The post-fracture viscous deformations are not in accord with field observations and rather represent a simplification of the larger-scale plastic regime. These points are clarified at L142-L143 of the revised manuscript.

4. Page 4: Equations 6 and 7: You are using multiple notations for the same thing: x and y components, 1 and 2 components. In Eq. 5, C is a fourth order tenor, Eq. 6 is a $2 \times 2$ matrix (components of a second order tensor?).

>> We changed the indices "1" and "2" for "x" and "y" in Eq. 7, in the revised manuscript, as suggested by the reviewer.

As the reviewer points out, the tensors defined in Eq. 5 are presented in Eq. 6 and 7 using a matrix notation. This notation is often used in the literature to concisely write the components of the elastic tensor, based on the symmetry of the stress and strain tensors. It is obtained by laying out the 3 (in 2D) independent components of the 2nd order stress and strain tensors into a single row, such that the components of the 4th order tensor C are written in a 3x3 matrix (6x6 in 3D). See Rice (2010) for reference. This is clarified at L126-128 of the revised manuscript.

5. Page 5: Probably helpful to define σI and σII in terms of stress components.

>> This is added as Eq. 11-12 in the revised manuscript, as suggested by the reviewer.

6. Page 6: Line 150: What is the 'standard' path?

>> The "standard" path refers to the original return algorithm in the EB and MEB damage parameterization (Rampal et al. 2016, Dansereau et al. 2016), where the super-critical stresses are relaxed along a line that runs through the origin. This is now specified in a new paragraph added at the end of section 2.4 in the revised manuscript.

7. Page 6: Line 151: Change 'to for' to 'for'.

>> Corrected as suggested by the reviewer.

8. Page 6: Line 162: Schreyer et al do not use 'granular theory', assuming that means models of granular flow. It is also confusing to refer to σc as a decohesive stress tensor since it has no apparent connection to Schreyer et al.

>> We remove "granular theory" in this sentence and use the term more carefully throughout the revised manuscript. We also clarified our references to Schreyer et al. (2006).

 Although our work is inspired by Schreyer et al. 2006, the name "decohesive stress tensor " is not a direct reference to their algorithm, but rather a reference to the fact that this stress is produced in association with the development of damage, hence to the decohesion of the ice material. We nonetheless note that our mention of Schreyer et al. 2006 refers to their use of a decohesive strain that is subtracted from the local elastic strain in the stress-strain relationship when the ice fractures, effectively relaxing the stress rates. This was clarified in the revised manuscript at L214-219, which now reads:

*"Note that the decohesive stress tensor used in this parameterization has a similar role as the decohesive strain used in the Elastic-Decohesive model (Schreyer et al., 2006). In Schreyer et al (2006), the decohesive strain represents the discontinuity in sea-ice displacement associated with a fracture and relaxes the effective stress rates. It is derived from a decohesion function that depends on the mode of failure. Here, we do not define the strain discontinuity associated with the fractures, but use the decohesive stress tensor $\boldsymbol{\sigma}_D$ to prescribe the orientation at which the stress state is relaxed back onto the yield curve. This only indirectly influences the local strain rate via the constitutive equation."*

9. Page 7: Line 177: Change 'correspond' to 'corresponding'.

>> Corrected as suggested by the reviewer.

10. Page 7: last line: What is included in the 'solution vector'?

>> The discretized set of equation corresponds to a system of N non-linear equations in the form of :

$$A x = b ,$$

where **x** is a vector formed by stacking all the $u_{i,j}$ components followed by the $v_{i,j}$ components, A is a NxN matrix with components that contains coefficients for the $u_{i,j}$ and $v_{i,j}$ dependent terms and B is a vector of length N containing the other terms. Then, the solution vector **F** is written as:

$$\mathbf{F} = \mathbf{A}\,\mathbf{x} - \mathbf{B}\,.$$

We chose not to include these details in the revised manuscript, as it would necessitate a lengthy numerical description, and added instead a reference to Lemieux et al., (2014) for readers in search of these precisions.

11. Page 8: Line 204: τa is a vector. I assume the scalar value you assign to it is for one component and the other is zero. (Also Page 9, Line 245.)

>> Yes. This precision is added at L256 in the revised manuscript.

12. Page 9: First line: Please give a reference showing the connection between failure in granular material and sea ice under uniaxial compression.

>> This section (4.3.3) and section 4.1 are re-written in the revised manuscript to clarify the granular character of sea ice and how it is related to fracture angles in uniaxial compression tests. A few references are added, such as Bardet et al. (1991) for granular geo-materials in uniaxial compression, Wachter et al. (2008) for fracture angles in ice samples, Overland et al (1998) for shear bands observations in the Arctic.

13. Page 9: Line 226: I don't see a definition of δ.

>> It is defined above Eq. 29, as the angle of dilatancy.

14. Page 9: Line 228: 'In general, the fracture angle ...' is this the fracture angle for sea ice?

>> We changed "In general" for "In most materials"

15. Page 9: Line 244: Change 'waves' to 'wave'.

>> Corrected as suggested by the reviewer.

16. Page 9: Line 248: '4 cfi)' means Figure 4?

>> This was an error in the labeling, and is corrected in the revised manuscript

17. Page 10: Line 254: Change 'are' to 'is'. I do not see in Fig. 5 that large values of R is associated with growth in εsym. Can you illustrate this better?

>> We do not expect a correlation between R, the largest local error in the damage factor Psi, and $\varepsilon_{sym}$, which represents the domain-integrated asymmetries in the stress field. Specifically, $\varepsilon_{sym}$ corresponds to the cumulated the far-field response to all residual errors of previous time iterates. It grows with the onset of fracture (with large R values), given that the damage parameterization is the largest source of errors. With time, however, the far-field response to previous errors become large and dominates over the new errors in the damage factor Psi. We clarified this point and these variables in section 4.3.1 and at L281-L285 in the revised manuscript.

18. Page 10: Line 255: Change 'growths R' to 'growth in R'.

>> For consistency, we changed for "error amplification ratio R", as used throughout the manuscript.

19. Page 10: Line 258: Change 'indicate' to 'indicates'.

>> Corrected as suggested by the reviewer.

20. Page 10: Line 269: Change 'depends on corrected' to 'depend on the corrected'.

>> Corrected as suggested by the reviewer.

21. Page 10: Lines 274-278: MEB and VP (and granular material models) make different predictions. Is there any evidence for your model behavior in experiments? The VP model is based on plasticity there is no fracture, so no 'post-fracture behavior.'

>> As discussed in comment #3 above, the large-scale sea-ice deformations are mostly "plastic" (see Coon et al., 1974, Tuhkuri and Lensu, 2002). The post-fracture viscous deformations in the MEB model thus do not correspond to field observations and represent a simplification of the plastic regime. The VP model simulates the plastic deformations associated with the ductile fractures, which corresponds to the observed material behaviour at the macro-scale. The VP model however does not represent the brittle component of the fractures or discontinuities in material properties, which occur at the smaller scales but may influence the fractures orientation and other deformation statistics. These points are clarified at L363-L366 in the revised manuscript.

22. Page 11: Line 284: Change 'approaches' to 'approach'.

>> Corrected as suggested by the reviewer.

23. Page 11: Line 286: Change 'sensitive other' to 'sensitive to other'.

>> Corrected as suggested by the reviewer.

24. Page 11: Line 290: Change 'increase' to 'increases'.

>> Corrected as suggested by the reviewer.

25. Page 12: Line 332: Change 'divergence' to 'divergent'.

>> Corrected as suggested by the reviewer.

26. Page 12: Line 335: Change 'reach' to 'reaches'. Change 'wave' to 'waves'.

>> Corrected as suggested by the reviewer.

27. Page 13: Line 357: Change 'generalizes' to 'generalized'.

>> Corrected as suggested by the reviewer.


June 21st, 2020

Note:

- The referee comments are shown in black,
- The authors answers are shown in blue,
- *Quoted texts from the revised manuscript are shown in italic and in dark blue.*
- Amendments made to the responses in the open discussion are shown in green

In this paper the authors introduce a modification of the MEB rheology in the form of a generalized damage parameterisation. They then proceed to test this new parameterisation using an idealised uniaxial loading setup. They find that the new parameterisation influences the resulting fracture angle, bringing it in the range of observations. The paper is well written and clear, using good English and sentence structure, and a logical flow from section to section and paragraph to paragraph.

The introduction of a modification of the MEB rheology is a niche topic, but potentially an important one and certainly one relevant for publication in the Cryosphere. As it stands, the paper has some faults I would like the authors to address. I expect they can do this adequately and that the resulting work will be fit for publication in the Cryosphere.

We thank the referee for his or her thorough review of the manuscript and constructive comments.

Major comments:

It is not clear why the authors are proposing this addition to the MEB. Is it numerics or physics, or something else? You say something general at the start, but it's vague and really only says what your modification does, not why you want to do it in the first place. This point should be crystal clear and guide the entire paper. Ideally the authors should say something like: "we want to introduce this scheme because we know it represents better the physics (and is incidentally better for the numerics). We see this by looking at the fracture angles (or some other measure)". Such a statement at the top would make this paper very strong.
An admittedly overly harsh evaluation of the current state is that the authors change something for dubious reasons and get a different response – so why should we care? Is this the right result, but for the wrong reasons? I don't think that's a fair assessment, but unless the motivation is clearer it will be the impression a critical reader gets.

>> We re-wrote the introduction to better state our objectives. The goal of the study is to reduce the integration errors in the MEB rheology and study the sensitivity of the model to the stress correction scheme, given that the exact path along which the super-critical stresses should be returned to the yield curve is not known a priori. This is clarified in the abstract, at L84-L90 and at L180-L188 in the revised manuscript.

We also add that our assessment of the sea ice deformations resulting from the use of a damage parameterization in the MEB model contributes to the current effort to assess the difference between different rheologies in reproducing satellite-derived sea-ice deformations (the FAMOS Sea-Ice Rheology Experiment (SIREx), https://epic.awi.de/id/eprint/48616/, with two papers currently under review in JGR). This is mentioned in the revised introduction.

Related to this lack of clear focus, I find it difficult to understand why you do the experiments that you do, so reading sections 4 and 5 is more demanding of the reader than it need be.

>> There is a clear need to standardized simple idealized experiments to test/evaluate different rheological models. This was identified at the workshop "Defining a cutting-edge future for sea-ice modelling" (Laugarvatn, Iceland, 2019) and again recently at the online workshop Modeling the Granular Nature of Sea Ice (https://seaicemuri.org/workshop.html). In both workshops, the simple uni-axial loading test (used in Ringeisen et al. (2019, 2020) received good acceptance for the community. This choice of experiment is clarified in section 4.1 of the revised paper.

I also question the fact that the authors don't introduce heterogeneity into their model. They even point out themselves that it "is responsible for much of the brittle material behaviour in progressive damage models" and indicate that the residual errors are not important in a heterogeneous field – which is what MEB is supposed to give. This choice needs to be much better justified than is currently done.

>> We did not include heterogeneity in order to clearly identify the model performance (both numerics and physics). The issues related to the error growth leading to asymmetry in a problem with full symmetry and their impact on the fracture angles could not be addressed using heterogeneity. This was clarified in the revised manuscript at L258-261. We also clarify at L438-441 in the revised manuscript that the heterogeneity is responsible for the localisation and intermittency of sea ice, properties that are not investigated in our manuscript.

Finally, there's almost a hostile tone towards the MEB rheology in the discussion and conclusion section. The authors are practically gleeful in pointing out various faults of the model that are not relevant to the modifications they propose. It is of course fine to point out the faults of MEB - which apparently are plentiful – but the way it is done here borders on un-professional, in my opinion.

>> This is a serious accusation (unprofessionalism). We would ask that the reviewer identify the offending sentences and we will respond promptly whether the paper is accepted or not.

Clearly, this is not our point of view. We disagree that our tone is hostile towards the MEB

model. We developed the only (to our knowledge) implementation of this rheology in a finite difference framework in order to be able to study the difference in physics independent of the numerics (other MEB implementations are done in Finite Element). Our study of the numerical and mechanical behaviour of this rheology is in the prospect of better understanding how the damage parameter simulates the deformations and to identify the key elements that can be useful to other models as we aim for higher resolution products. The current paper is a follow-up to an earlier paper where those issues were raised but not addressed. Our goal is to improve sea ice modeling in general and we believe that a multi-model approach towards this goal is very useful.

Minor comments:

L16: The formulation makes it sound as if leads and LKFs are interchangeable, but they are not.

>> We agree and removed to mention to LKF in this sentence.

L120: Shouldn't the cohesion be a function of resolution (see Weiss, 2007)? If that's the case, how do you get the same value from large scale and the lab?

>> The material strength is a function of the resolution, and we do expect smaller values at the large scales (kms in our model) than in laboratory experiments, which usually find strengths that are one or more orders of magnitude larger than what we use in our study (10 kN m$^{-2}$). Our choice of cohesion is based on results from the ice bridge experiments of Plante et al. 2020, and coherent with what has been used in other studies using the MEB rheology (e.g. Dansereau et al. 2016, 2017, 2019, and also in Rampal et al. 2016, 2019).

L135: What's the physical justification for proposing this generalised stress correction?

>> As we mentioned above, we develop the generalised stress correction in part to improve the issues identified in our previous paper, and in part to assess the influence of the super-critical stress correction on the simulated fractures and deformation, with minimal chances to the damage parameterization. This is clarified in the revised manuscript at L84-90 and at the beginning of section 3.

Note that in the original parameterization, the choice of defining the damage parameter in terms of the amount of stress in excess of the yield curve was made to offer numerical robustness and simplicity. In a perfect model for instance, this overshoot would approach zero. A physically meaningful definition of the damage parameter could involve thermodynamics relations as the stress state approaches the yield curve (see for instance Murakami 2012), or use discrete cycling methods (as in the models of Main., 2000, Amitrano and Helmstetter., 2006, Carrier et al., 2015), but would represent a significant modification of the damage parameterization. This is considered for future model development but out of the scope of this paper.

L222: Mohr-Coulomb and Roscoe theories both concern granular materials, but hear we're dealing with the fracturing of a solid. Are they still valid? Please elaborate.

>> Sea ice is a granular material. Sea for instance books from Leppäranta (2011), Weiss (2013),

and the recent workshop "Modeling the Granular Nature of Sea Ice" (https://seaicemuri.org/workshop.html), bringing scientists from all around the world working on this topic, for reference.

L248: There's a lot of information in figure 4 and the reader needs more help in deducing why you created it and what it's supposed to tell us.

>> Additional information is included in the figure caption of the revised manuscript, which now reads:

*"Scatter plots of local stress invariants ($\sigma_{I}$ vs. $\sigma_{II}$, in kN m$^{-1}$, left column),  normal stresses and scaled strain rate invariants ($\sigma_{I}$ vs. $(1-d)^3 \dot{\epsilon}_{II}$, right column) in heavily damaged (d > 0.9) grid cells, at t = 57 min (during the fracture development, top row), t = 60 min (a few minutes after the fracture, middle row), and t = 90 min ($\sim$ 30 min after the fracture, bottom row). Color indicates the local damage. The strain rates are normalised to account for the non-linear dependency of the viscosity $\eta$ on the damage parameter. The gradual alignment of the points in the $\sigma_{I}$ vs. $(1-d)^3 \dot{\epsilon}_{II}$ diagram indicate the development of a linear-viscous stress-strain relationship over time."*

L276: A reference to the contrasting results is needed.

>> We add the reference to Ringeisen et al. (2019) for the VP model and to Bardet (1991), Balendran and Nemat-Nasser (1993), for granular materials.

L291: A reference for what is typical for granular material is needed (a textbook will suffice).

>> We added areference to the Bolton  et al. (1986), describing the sawtooth model of granular dilatancy.

L316: This entire paragraph is a bit up-side-down to me. You start by saying the MEB is not good enough, for various reasons (begging the question of why you use it in the first place, actually) - and then you say how your new addition will not save it. A more natural way to write this is to first say that although the decohesive stress tensor can do some things it cannot fix everything, including etc.

>> We do not state that the MEB is not good enough. We mention the differences in behaviour with respect to the more commonly used VP models and discuss these differences in terms of potential limitations that should be taken into account in future model developments. We do believe that the use of several different models raises questions that would not be raised with the use of a single model, even if that model is better. Our community has suffered from a monopoly in approach with the standard VP model, until only very recently when new approaches were developed. This is made clear in the introduction of the revised manuscript.

We also believe that this discussion is made clearer in the revised manuscript by specifying in the abstract, at L84-90 and L180-L188 that we developed the generalized damage parameterization

in part to investigate the influence of the return algorithm on the simulated fractures.

L329: This paragraph is off topic, discussing experiments not introduced before and not relevant to the introduction of the decohesive stress tensor. Please remove.

>> We argue that this paragraph serves to put our results in context with other MEB model studies, which use different material parameters. In the revised manuscript, we widen the discussion to integrate the effect of grid resolution, sample aspect ratio, advection and heterogeneity, and clarify the context of this discussion.

L353: Now I'm confused, did you want to solve the ridging problem by introducing the decohesive stress? Again, a more natural way to present your results would be to first state what works and then what remains.

>> We now specify that the generalized damage parameterization modification is used to tackle issues that we raised on the damage parameterization in a previous paper (Plante et al. 2020) but also to investigate the influence of the return algorithm on the simulated fractures. The dominance of the post-fracture deformations in the MEB rheology is an important finding in our experiments, which contrasts with the behaviour in the VP and EVP rheologies. This is clarified in the conclusion at L443-449, but also in the abstract, at L84-L90 and at L180-L188 in the revised manuscript.

Note:

- The referee comments are shown in black,
- The authors answers are shown in blue,
- *Quoted texts from the revised manuscript are shown in italic and in dark blue.*
- Amendments made to the responses in the open discussion are shown in green

Please consider this an amendment to my review, RC3.

I now realise that the paper is fundamentally flawed and should be rejected/withdrawn.

The authors propose that super-critical stress be relaxed onto the failure envelope via a path different from the shortest distance to the origin in ($\sigma_I$, $\sigma_{II}$) space - as per figure 1. In order to do so they propose calculating the damage factor $\Psi$ via equation (17) or (25, which has a typo). This, however, will not have the desired effect, because $\Psi$ scales $d$, which then scales *all* the components of $\sigma$ *equally*. Please consider equations (12), (8), (9), and (5) to see how changing $d$ affects $\sigma$.

In other words, as long as $d$ is a scalar, and not a tensor, then any form of $\Psi$ will reduce the stress towards the origin - $\Psi$ only determines how fast this happens.

For a more concrete example consider the stress change $\sigma'$ -> $\sigma_c$ in figure 1a. In order to change the stress in this manner, we need to reduce the shear stress but *increase* (in absolute value) the normal stress. This cannot be done by increasing $d$.

A generalised stress correction scheme, therefore, requires $d$ to be a tensor, so that different components of the stress can change differently. This is not done here, and the scheme proposed simply does not do what the authors claim it does. Instead of taking a different path to the failure envelope, the proposed scheme takes the same path as the standard scheme but increases the damage too little for the stress to cease being super-critical. The proposed scheme is thus not usable and a paper discussing it is not warranted.

\>> It is correct that the damage factor Psi is a scalar, but the reviewer missed the fact that we use a decohesive stress tensor to bring the stress back onto the yield curve, and which depends on the correction path angle (see Eqs 18-20, discussion on L150-161, Fig 1, and derivation below). This is emphasized in the revised manuscript at L204-207.

Therefore, the corrected normal stress invariant reads:

$\sigma_{Ic} = \Psi \sigma'_I + \sigma_{ID},$

rather then $\sigma_{Ic} = \Psi \sigma'_I$ , which was the reviewer's concern.

The components of the decohesive stress tensor are then retrieved using the yield criterion, such that:

$$\frac{c - \sigma_{IIc}}{\mu} = \Psi \sigma'_I + \sigma_{ID}$$

$$\sigma_{ID} = \frac{c - \sigma_{IIc}}{\mu} - \Psi \sigma'_I$$

Using the relation $\sigma_{IIc} = \Psi \sigma'_{II}$, we get Eq. 19 (where there was a typo):

$$\sigma_{ID} = \frac{c - \Psi(\sigma'_{II} + \mu \sigma'_I)}{\mu}$$

In which $\Psi$ depends on the correction path angle. We can provide a more complete derivation in this thread if desired.

[revised manuscript text omitted]

---

## Author Response (AR2)

**Authors answers to tc-2020-354 comments**

**November 5ᵗʰ, 2021**

Dear Editor,

Thank you for accepting our manuscript for publication. We are pleased to resubmit a corrected version of our revised manuscript entitled "A generalized stress correction scheme for the MEB rheology: impact on the fracture angles and deformations" by Mathieu Plante and L. Bruno Tremblay.

We would like to thank the reviewer for their useful comments and suggestions. We have addressed all comments included in the editor decision and corrected the manuscript accordingly. You will find below our specific responses to each of the comments.

Sincerely,

On behalf of all the authors,
Mathieu Plante

Note:

- The referee comments are shown in black,
- The authors' answers are shown in blue.

**Answers to tc-2020-354 comments**

November 5[th], 2021

Note:

- The referee comments are shown in black,
- The authors' answers are shown in blue,

Second review of "A generalized stress correction scheme for the MEB rheology: impact on the fracture angles and deformations" by Plante and Tremblay

The author adequately replied to all of my comments, in particular, my main concern of general introduction and problem statement has been addressed. During rereading the revised manuscript I noted a few typos and made a few suggestions, listed below.

>> We thank the referee for his or her thorough review of the manuscript and correction suggestions.

Typos/suggestions:

page 1
l3 suggestion: The conventional MEB
>> Corrected in the revised manuscript.

l12: longer-term without hyphen? better: in the long term
>> Corrected for "in the long term" in the revised manuscript.

page 2
l32 scheme -> schemes
>> Corrected in the revised manuscript.

l39: framework -> frameworks, please check the sentence, what does "that influence" refer to? Diversity, sea-ice theologies?
>> The typo is corrected and the sentence is clarified.

l48: produces -> produce
>> Corrected in the revised manuscript.

page 3
l67: have -> has

>> Corrected in the revised manuscript.

l72: suggestion: other neXtSIM specific characteristics, such as the Lagrangian numerical scheme,
>> We added their use of a triangular adaptive mesh.

page 6
l150: eq: 12, should this be \left(\frac\sigma_{xx}-\sigma_{yy}}{2}\right)^2
>> Yes, this is corrected in the revised manuscript.

page 9
l235: shouldn't that be \sigma_{I}' < 0?
>> Yes, this is corrected in the revised manuscript.

l249: (Ringeisen et al., 2019) fix parentheses
>> Corrected in the revised manuscript.

page 10
l268: "of the solution vector", isn't is the residual vector?
>> This is changed to "of the solution residual vector".

l279: (nx,ny) really picky, but I would still write $(n_x,n_y)$ one $(i,j)$ in the text.
>> We agree. This is corrected in the revised manuscript.

page 11
l290: analog -> analogous
>> Corrected in the revised manuscript.

page 12
l355: I think you mean this: amplification ratio R (see Eq. 27) that reaches ~20 in the control simulation (Fig. 5c).
>> Yes, we do mean the amplification ratio R. However, we keep "up to two orders of magnitude" as this comment also refers to the simulations with gamma<0. While the spikes in the control simulation only reach ~20, the exact magnitude of these dependent on simulation-specific residual errors.

We added the references to Eq. 27 and to Fig. 6b.

page 15
l420: increase->increased?
>> Corrected in the revised manuscript.

Section 7 conclusion with upper case C
>> Corrected in the revised manuscript.

[revised manuscript text omitted]